# Active Learning with Selective Time-Step Acquisition for PDEs

Yegon Kim [1]   Hyunsu Kim [1]   Gyeonghoon Ko [1]   Juho Lee [1]

## Abstract

Accurately solving partial differential equations (PDEs) is critical to understanding complex scientific and engineering phenomena, yet traditional numerical solvers are computationally expensive. Surrogate models offer a more efficient alternative, but their development is hindered by the cost of generating sufficient training data from numerical solvers. In this paper, we present a novel framework for active learning (AL) in PDE surrogate modeling that reduces this cost. Unlike the existing AL methods for PDEs that always acquire entire PDE trajectories, our approach strategically generates only the most important time steps with the numerical solver, while employing the surrogate model to approximate the remaining steps. This dramatically reduces the cost incurred by each trajectory and thus allows the active learning algorithm to try out a more diverse set of trajectories given the same budget. To accommodate this novel framework, we develop an acquisition function that estimates the utility of a set of time steps by approximating its resulting variance reduction. We demonstrate the effectiveness of our method on several benchmark PDEs, including the Burgers' equation, Korteweg–De Vries equation, Kuramoto–Sivashinsky equation, the incompressible Navier-Stokes equation, and the compressible Navier-Stokes equation. Experiments show that our approach improves performance by large margins over the best existing method. Our method not only reduces average error but also the 99%, 95%, and 50% quantiles of error, which is rare for an AL algorithm. All in all, our approach offers a data-efficient solution to surrogate modeling for PDEs.

## 1. Introduction

In many scientific and engineering applications, accurately solving partial differential equations (PDEs) in the form of trajectories of states evolving over time is essential for understanding complex phenomena (Holton and Hakim, 2013; Atkins et al., 2023; Murray, 2007; Wilmott et al., 1995). The traditional approach involves running numerical solvers, which provide accurate solutions but are computationally costly, taking several hours, days or even weeks to run depending on the complexity of the problem (Cleaver et al., 2016; Cowan et al., 2001). As a result, there is significant interest in developing surrogate models (Greydanus et al., 2019; Bar-Sinai et al., 2019; Sanchez-Gonzalez et al., 2020; Z. Li et al., 2020; Brandstetter et al., 2022b; Lippe et al., 2024) that can approximate the solutions more efficiently. Surrogate models are obtained by solving regression tasks on some "ground truth" data. The ground truth data for PDEs are generated by numerical solvers, which are costly compared to those of standard regression problems. As a result, the expense of data acquisition presents a major bottleneck in the development of surrogate models for PDEs.

Active Learning (AL, Chernoff, 1959; MacKay, 1992; Settles, 2009) can address this challenge by adaptively acquiring the most informative inputs, effectively reducing the amount of ground-truth data required to obtain a high-quality surrogate model. However, there is a general lack of research in AL for regression tasks (Dongrui Wu, 2018; Holzmüller et al., 2023), let alone PDEs. Existing studies on AL for PDEs have predominantly dealt with univariate outputs such as energy (Pestourie et al., 2020; Pickering et al., 2022), or predictions at a single, fixed time point (Bajracharya et al., 2024; Dongxia Wu et al., 2023). To our surprise, the only work directly addressing AL for prediction of trajectories is that of Musekamp et al. (2024). In this work, the surrogate model is set as an autoregressive model that predicts the evolved state of a PDE at time $t + \Delta t$ given a state at an arbitrary time point $t$, and is trained on data acquired by existing regression-based AL methods (Holzmüller et al., 2023). Specifically, at each round of acquisition, the AL method chooses initial conditions from which *entire* trajectories are acquired. However, we argue that querying all the states in a trajectory is not sample-efficient, especially for autoregressive surro-

[1] Korea Advanced Institute of Science and Technology, Daejeon, Korea. Correspondence to: Yegon Kim <yegonkim@kaist.ac.kr>, Juho Lee <juholee@kaist.ac.kr>.

*Proceedings of the 42nd International Conference on Machine Learning*, Vancouver, Canada. PMLR 267, 2025. Copyright 2025 by the author(s).

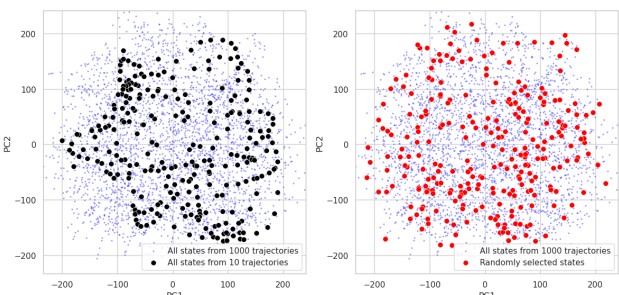

Figure 1: PCA of surrogate model hidden layer's activation patterns on states of the incompressible Navier-Stokes equation. The left figure highlights states within 10 trajectories, and the right figure highlights the same number of states chosen randomly.

gate models.

Acquiring entire trajectories is inefficient mainly for two reasons. First, states within a trajectory are often strongly correlated, undermining their diversity or the joint information gain (Houlsby et al., 2011; Kirsch et al., 2019). Fig. 1 shows a side-by-side comparison of PDE states selected within a few trajectories and the same number of states selected randomly from many trajectories. States within trajectories exhibit clustering and hence lower diversity compared to the randomly selected ones. Further analysis and details are given in Appendix D.2. Secondly, even if the states are not strongly correlated, it can be the case that only certain time steps of a trajectory are the most informative due to the dynamics of the PDE. In both cases, noting that the main cost is in running the numerical solver, it would be ideal to selectively acquire only the most important time steps with the numerical solver, for a fraction of the cost of acquiring the entire trajectory. However, this is usually impossible without querying all the time steps that come earlier.

In this paper, we propose a novel framework for data acquisition that circumvents the constraint of having to query all time steps in a trajectory, along with an AL strategy that leverages this flexibility. Our method combines both a numerical solver and a surrogate model to acquire data along a trajectory with reduced cost. Specifically, it selects which time steps along a trajectory to query to the solver, while using the surrogate model to approximate the remaining steps. To do so, we develop a novel acquisition function that guides our AL strategy in choosing which time steps to query to the numerical solver in each trajectory.

Overall, our framework, equipped with the novel AL strategy, significantly improves surrogate model performance over previous methods. We validate our approach through extensive experiments on benchmarks, including the Burgers' equation, the Korteweg–De Vries equation, the Ku-

ramoto–Sivashinsky equation, the incompressible Navier-Stokes equation and the compressible Navier-Stokes equation. The compressible Navier-Stokes equation is a particularly challenging task due to its high nonlinearity and turbulent behavior. Additionally, we analyze the behavior of our AL method, providing insights into the factors that contribute to its effectiveness. Our code is publicly available at `https://github.com/yegonkim/stap`.

## 2. Background

### 2.1. Preliminaries

We consider PDEs with one time dimension $t \in [0, T]$ and possibly multiple spatial dimensions $\boldsymbol{x} = [x_1, x_2, \ldots, x_D] \in \mathbb{X}$ where $\mathbb{X}$ is the spatial domain such as the unit interval. These can be written in the form

$$\partial_t \boldsymbol{u} = F(t, \boldsymbol{x}, \boldsymbol{u}, \partial_{\boldsymbol{x}} \boldsymbol{u}, \partial_{\boldsymbol{xx}} \boldsymbol{u}, \ldots), \qquad (1)$$

where $\boldsymbol{u} : [0, T] \times \mathbb{X} \to \mathbb{R}^n$ is a solution to the PDE. We are also given a specific boundary condition and a fixed time interval $\Delta t$. If the PDE is well-posed (W. Evans, 1988), there exists, for each $t_0 \in \mathbb{R}$, an evolution operator $G_{t_0}$ which maps an initial condition $\boldsymbol{u}^0 := \boldsymbol{u}(t_0, \cdot)$ to the solution $\boldsymbol{u}^1 := \boldsymbol{u}(t_0 + \Delta t, \cdot)$. For simplicity, we only consider time-independent PDEs, for which the evolution operator $G_t$ is the same for all $t$, say $G$. Iterating over $G$ multiple times, we can obtain a trajectory $(\boldsymbol{u}^i)_{i=1}^{L}$ of length $L$, where $\boldsymbol{u}^i := G^{(i)}[\boldsymbol{u}^0]$ with $G^{(i)}$ being the $i$-th iterate of $G$. In practice, $G$ is implemented by various numerical methods, adequately chosen for the given PDE, as elaborated in Appendix B.1.

We train a neural surrogate model $\hat{G}$ with input-output pairs $(\boldsymbol{u}, G[\boldsymbol{u}])$ from the numerical solver $G$. Active learning aims to build a high quality training dataset by adaptively selecting informative inputs to be fed into the solver $G$. Prior work (Musekamp et al., 2024) operates on the the framework where initial conditions $\boldsymbol{u}^0$ are selected from a pool $\mathcal{P}$, from which full trajectories of length $L$ are obtained. For instance, Query-by-Committee (**QbC**, Seung et al., 1992) queries initial conditions $\boldsymbol{u}^0$ that maximize the predictive uncertainty estimated from a committee of $M$ models,

$$a_{\text{QbC}}(\boldsymbol{u}^0) = \frac{1}{M} \sum_{m=1}^{M} \sum_{i=1}^{L} \|\hat{\boldsymbol{u}}_m^i - \bar{\boldsymbol{u}}^i\|_2^2 \qquad (2)$$

where $\hat{\boldsymbol{u}}_m^i$ is the prediction of the $i^{\text{th}}$ state from the $m^{\text{th}}$ surrogate model in the committee and $\bar{\boldsymbol{u}}^i := \frac{1}{M} \sum_{m=1}^{M} \hat{\boldsymbol{u}}_m^i$ is the mean prediction from the committee.

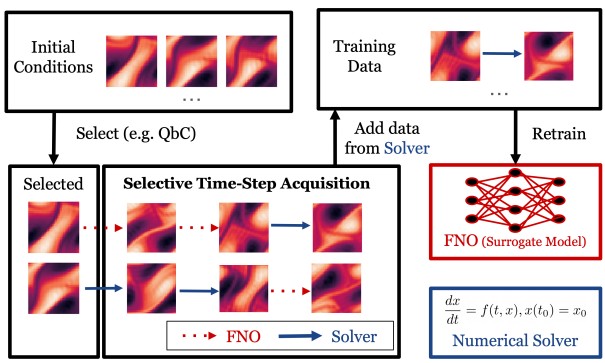

Figure 2: Illustrated overview of STAP. This illustration describes one round of AL.

## 2.2. Problem Setting

Our goal is to obtain a surrogate model $\hat{G}$ that approximates the expensive numerical solver $G$ with low error

$$\mathbb{E}_{\boldsymbol{u}^0 \sim p(\boldsymbol{u}^0)} \left[ \text{err} \left( (G^{(i)}[\boldsymbol{u}^0])_{i=1}^L, (\hat{G}^{(i)}[\boldsymbol{u}^0])_{i=1}^L \right) \right] \quad (3)$$

where $\text{err}(\cdot, \cdot)$ is an error metric. Obtaining the surrogate model requires sampling training data from the numerical solver, which incurs a nontrivial cost. AL aims to improve sample efficiency by sampling only the most important data. In particular, AL utilizes the current surrogate model $\hat{G}$, or a committee of surrogate models $\{\hat{G}_m\}_{m=1}^M$, to inform its choice. After acquiring the data chosen by AL, the surrogate $\hat{G}$ is retrained with the new dataset.

We assume that there exists a pool $\mathcal{P}$ of initial conditions $\boldsymbol{u}^0$. At each round of AL, we train a committee of $M$ surrogate models $\{\hat{G}_m\}_{m=1}^M$ with the training dataset collected from $G$ up to that round of AL. We then use this committee to select a batch of inputs to be queried to the solver $G$, and add the pairs of queried inputs and resulting outputs to the training dataset. The cost at each round, defined as the number of inputs queried to $G$, is limited to a certain budget $B$. We aim to achieve low errors at each round, so an AL strategy would ideally acquire data with cost as close to or equal to the budget (S. Li et al., 2022a).

## 3. Selective Time-Step Acquisition for PDEs

### 3.1. Framework of Data Acquisition

We present our method, **S**elective **T**ime-Step **A**cquisition for **P**DEs (STAP), which operates under a framework of data acquisition that is much more sample efficient than previous works. Algorithm 1 provides an overview of our framework. Fig. 2 also provides an illustrated version of the overview. We start with a surrogate model $\hat{G}$, or a committee of surrogate models $\{\hat{G}_m\}_{m=1}^M$, trained with the initial dataset $\mathcal{D}$. At every round of AL, we choose an initial con-

---

**Algorithm 1** Overview of STAP

**Require:** Pool $\mathcal{P}$ of initial conditions, budget $B$ per round, number of rounds $R$, numerical solver $G$, trajectory length $L$, initial training dataset $\mathcal{D}$

**Ensure:** Trained surrogate model $\hat{G}$

1: Train $\hat{G}$ on $\mathcal{D}$
2: **for** round $= 1$ to $R$ **do**
3:      cost $\leftarrow 0$
4:      **while** cost $< B$ **do**
5:          Choose initial condition $\boldsymbol{u}^0$ from $\mathcal{P}$      ▷ § 3.3
6:          $\mathcal{P} \leftarrow \mathcal{P} \setminus \{\boldsymbol{u}^0\}$
7:          Choose $S = (s_1, \ldots, s_L)$      ▷ §§ 3.2 and 3.3
8:          $\hat{\boldsymbol{u}}^0 \leftarrow \boldsymbol{u}^0$
9:          **for** $i = 1$ to $L$ **do**
10:             $\hat{\boldsymbol{u}}^i \leftarrow \begin{cases} G[\hat{\boldsymbol{u}}^{i-1}] & \text{if } s_i = \text{true} \\ \hat{G}[\hat{\boldsymbol{u}}^{i-1}] & \text{if } s_i = \text{false} \end{cases}$
11:             **if** $s_i = \text{true}$ **then**
12:                 $\mathcal{D} \leftarrow \mathcal{D} \cup \{(\hat{\boldsymbol{u}}^{i-1}, \hat{\boldsymbol{u}}^i)\}$
13:                 cost $\leftarrow$ cost $+ 1$
14:             **end if**
15:          **end for**
16:      **end while**
17:      Train $\hat{G}$ on $\mathcal{D}$
18: **end for**

---

dition $\boldsymbol{u}^0$ from the pool $\mathcal{P}$, similar to the existing AL methods for PDE trajectories. However, while existing methods acquire the entire trajectory starting from the chosen initial condition $\boldsymbol{u}^0$ (Musekamp et al., 2024), our method acquires a partial trajectory. Specifically, we select a subset of time steps to simulate from $\boldsymbol{u}^0$, rather than acquiring the full trajectory. The rationale behind this approach is that, given a fixed budget, acquiring as many trajectories as possible—albeit partially—from different initial conditions is often more beneficial than fully acquiring fewer trajectories. This strategy enables more efficient exploration of the data space and improves the overall sample efficiency of the framework.

More specifically, for a given initial condition $\boldsymbol{u}^0$, we define a boolean sequence of length $L$, $S = (s_1, \ldots, s_L)$, which we refer to as the *sampling pattern*. For example, $S$ could be $(\text{true}, \text{false}, \ldots, \text{true})$. The sampling pattern specifies that data will be acquired only at time steps corresponding to true values while skipping those marked false.

After selecting the sampling pattern $S$, the next step is to acquire the PDE trajectory. While acquiring a full trajectory is straightforward using a numerical solver $G$, obtaining a partial trajectory corresponding to $S$ can be tricky. We want to run the solver $G$ only for the time steps specified by $S$ (those with true patterns), but the solver requires the skipped time steps (those with false patterns) as intermediate inputs. If we give up and just run the solver for all time steps for this reason, we wouldn't be saving any cost. To address this, we use a simple heuristic: for the skipped time steps, we replace the simulation with predic-

tions from the *surrogate model* (we use the average surrogate $\hat{G} = \frac{1}{M} \sum_{m=1}^{M} \hat{G}_m$ when we have a committee). That is, starting with $\hat{\boldsymbol{u}}^0 = \boldsymbol{u}^0$, we iterate over $1 \leq i \leq L$:

$$\hat{\boldsymbol{u}}^i = \begin{cases} G[\hat{\boldsymbol{u}}^{i-1}] & \text{if } s_i = \text{true} \\ \hat{G}[\hat{\boldsymbol{u}}^{i-1}] & \text{if } s_i = \text{false.} \end{cases} \quad (4)$$

We add to our dataset $\mathcal{D}$ only the pairs obtained with the solver $G$, namely $(\hat{\boldsymbol{u}}^{i-1}, \hat{\boldsymbol{u}}^i)$ with $s_i = \text{true}$.

In comparison to full trajectory acquisition, which requires $L$ executions of the numerical solver, our strategy invokes the numerical solver $\|S\| := \sum_{i=1}^{L} \mathbb{1}[s_i = \text{true}]$ times and utilizes the surrogate model $L - \|S\|$ times. Since the surrogate model is significantly cheaper to run than the numerical solver, this approach substantially reduces the cost of acquisition, enabling us to explore more initial conditions within the same budget. In fact, as discussed in § 2.2, we define the acquisition cost precisely as $\|S\|$. We repeat expanding our training dataset with new initial conditions and sampling patterns until the cost incurred in the current round reaches a budget $B$. At the end of each round, we retrain the surrogate $\hat{G}$ with the expanded training set $\mathcal{D}$.

Previous methods listed in Musekamp et al. (2024) can be considered a special case of ours where the sampling pattern $S$ is full of true entries. Our framework is therefore a strict generalization of previous works. In the remainder of this section, we describe how STAP adaptively chooses initial conditions $\boldsymbol{u}^0$ and sampling patterns $S$.

### 3.2. Acquisition Function

To adaptively select the sampling pattern $S$ with the initial condition $\boldsymbol{u}^0$, we propose a novel acquisition function $a(\boldsymbol{u}^0, S)$ that assesses the utility of $S$. Given a committee $\{\hat{G}_m\}_{m=1}^M$, consider $(\hat{G}_a, \hat{G}_b)$ for some $a \neq b \in \{1, \ldots, M\}$. We define the utility of the sampling pattern $S$ for the pair $(\hat{G}_a, \hat{G}_b)$ as the resulting *variance reduction* in the pair's rolled-out trajectories. Specifically, let $\hat{\boldsymbol{u}}_a$ and $\hat{\boldsymbol{u}}_b$ be the trajectories estimated by $\hat{G}_a$ and $\hat{G}_b$, starting from $\boldsymbol{u}^0$. Next, let $\hat{\boldsymbol{u}}_{b,S,a}$ be the trajectory rolled-out using Eq. 4 but with $\hat{G}$ and $G$ replaced by $\hat{G}_b$ and $\hat{G}_a$, respectively. In other words, $\hat{G}_b$ is used at time steps where the sampling pattern indicates false and $\hat{G}_a$ otherwise. This trajectory is designed to approximate $\hat{G}_b$ that's updated itself with data from $\hat{G}_a$ at time steps for which the sampling pattern indicates true. The variance reduction is defined as

$$R(a, b, S) := \sum_{i=1}^{L} \left( \|\hat{\boldsymbol{u}}_a^i - \hat{\boldsymbol{u}}_b^i\|^2 - \|\hat{\boldsymbol{u}}_a^i - \hat{\boldsymbol{u}}_{b,S,a}^i\|^2 \right). \quad (5)$$

The sampling pattern $S$ that maximizes $R(a, b, S)$ is the one where the current models $\hat{G}_a$ and $\hat{G}_b$ disagree the most, and acquiring data from $S$ effectively reduces this discrepancy. Our acquisition function is defined as the average

variance reduction between all the distinct pairs in the committee:

$$a(\boldsymbol{u}^0, S) = \frac{1}{M(M-1)} \sum_{a,b \in \{1, \cdots, M\}, a \neq b} R(a, b, S). \quad (6)$$

We observe that our acquisition function simplifies to QbC in Eq. 2 when $S$ acquires all the time steps, differing only by a constant factor of two. This occurs because, in that case, $\hat{\boldsymbol{u}}_{b,S,a} = \hat{\boldsymbol{u}}_a$, which makes the second term in the summand of Eq. 5 vanish. Consequently, we can interpret our acquisition function as a generalization of QbC that accommodates for the selection of time steps.

As an additional sanity check, consider the scenario where $S$ does not sample any time steps. In this situation, $\hat{\boldsymbol{u}}_{b,S,a} = \hat{\boldsymbol{u}}_b$, leading the two terms in the summand to cancel each other out, resulting in zero variance reduction. Since acquiring no data should yield zero utility, we confirm that our acquisition function behaves as expected in this limiting case. Appendix A.2 further details the precise motivation behind the design of our acquisition function.

### 3.3. Batch Acquisition Algorithm

Since retraining on a new dataset with every new data point is costly, and acquiring data in a parallel manner can be more efficient than when done in a sequential manner, active learning algorithms should accommodate batch acquisition, that is, the acquisition of multiple data points at each round of active learning (Kirsch et al., 2019). There are two main challenges in designing a batch acquisition AL methods. First, simply maximizing the acquisition values of individual data points doesn't maximize the utility of the batch. Numerous works report that picking instances that maximize individual acquisition values can severely underperform compared to methods that take into account the interactions between those instances (Kirsch et al., 2019; Ash et al., 2019). The problem is chiefly attributed to the lack of diversity and representativeness (Dongrui Wu, 2018) caused by oversampling of small, high value regions (Smith et al., 2023). Secondly, the AL method needs to search over a large combinatorial space of size $O(|\mathcal{P}|^B)$, where $|\mathcal{P}|$ and $B$ are the pool size and batch size, respectively. The search method should therefore be scalable. Most existing batch AL algorithms address both issues (Holzmüller et al., 2023).

With the acquisition function defined in § 3.2, we present a batch acquisition algorithm given a pool $\mathcal{P}$ of initial conditions. We define the cost of a batch $\{(\boldsymbol{u}_j^0, S_j)\}_{j=1}^N$ as the total number of queries to the solver, $\sum_{j=1}^N \|S_j\|$. If we were to maximize the sum of individual acquisition values $\sum_{j=1}^N a(\boldsymbol{u}_j^0, S_j)$ under a budget constraint $\sum_j \|S_j\| \leq B$, there is a known approximate solution Salkin and De

Kluyver (1975) that greedily maximizes the cost-weighted acquisition function $a^*(\boldsymbol{u}^0, S) = a(\boldsymbol{u}^0, S)/\|S\|$ until the total cost exceeds the budget $B$. However, as discussed above, it's questionable whether the sum of individual acquisition values is actually a good representative for the utility of a batch. Moreover, even if we employ the approximate greedy solution, we are still searching over the *product* pool of the sampling pattern $S$ and the pool $\mathcal{P}$ of initial conditions, whose size is on the order of $O(2^L|\mathcal{P}|)$. Both terms impose significant computational burden on optimizing the cost-weighted objective $a^*(\boldsymbol{u}^0, S)$.

We therefore propose STAP as an add-on to existing AL methods that acquire full trajectories. By using STAP as an add-on, the diversity and representativeness promoted by existing AL methods (Holzmüller et al., 2023; Kirsch et al., 2023; Musekamp et al., 2024) are upheld, and the size of the optimization space for STAP is reduced to $O(2^L)$. Specifically, a full-trajectory AL method $\mathcal{A}$, which we call a *base* method, first selects an initial condition $\boldsymbol{u}^0$. Musekamp et al. (2024) introduces several possibilities for such a method, including **QbC** (Seung et al., 1992), Largest Cluster Maximum Distance (**LCMD**, Holzmüller et al., 2023), **Core-Set** (Sener and Savarese, 2017), and stochastic batch active learning (**SBAL**, Kirsch et al., 2023). We then optimize the cost-weighted acquisition function $a^*(\boldsymbol{u}^0, S)$ over the sampling pattern $S$ while holding $\boldsymbol{u}^0$ fixed, and add the pair $(\boldsymbol{u}^0, S)$ to the current batch. We iterate this two-stage process until the cost of the batch reaches our budget $B$. Additionally, if the cost ever exceeds the budget after adding a pair, we truncate the sampling pattern so that the cost is exactly equal to the budget.

We use a greedy algorithm to optimize the cost-weighted acquisition function $a^*(\boldsymbol{u}^0, S)$ over $S$ in the combinatorial space of size $O(2^L)$. In the greedy algorithm, we start by initializing $S$ with all entries set to true. At each step of the greedy algorithm, we propose a neighboring pattern $S'$ by applying a bit-flip mutation, where each bit of $S$ is flipped with a probability of $\epsilon$. The proposal is accepted only if the acquisition value $a^*(\boldsymbol{u}^0, S')$ is higher than the current value $a^*(\boldsymbol{u}^0, S)$. This process of proposal and acceptance/rejection is repeated $T$ times. We use $T = 100$ and $\epsilon = 0.1$ throughout our experiments. A more concise summary of the batch acquisition algorithm is given in Appendix A.3. The algorithmic complexity of batch acquisition is discussed in § 5.6.

## 4. Related Work

**AL for PDEs.** The works by Pestourie et al. (2020), Pickering et al. (2022), and Gajjar et al. (2022) apply active learning to problems involving PDEs, but their tasks are limited to predicting a particular quantity of interest, such as the maximum value of an evolved state. S. Li et al. (2024)

and Dongxia Wu et al. (2023) apply their AL methods to single-state prediction. Bajracharya et al. (2024) explores the use of active learning in tasks of predicting steady states of PDEs, which can be seen as predicting single states at $t \to \infty$. Finally, Musekamp et al. (2024) experiments with active learning in predicting PDE trajectories with autoregressive models.

**Active selection of time points.** While our work is the first to propose time step selection in active learning (AL) for PDEs, the concept of selecting time points has been explored in other contexts. For example, in physics-informed neural networks (PINNs), active selection of collocation points for training has been widely studied (Arthurs and King, 2021; Gao and Wang, 2023; Mao and Meng, 2023; C. Wu et al., 2023; Turinici, 2024). "Labels" for PINNs, or the residual loss, can be calculated directly at any time point using closed form equations. There are also methods in Bayesian experimental design (BED) that choose observation times that maximize information gain about parameters of interest (Singh et al., 2005; Cook et al., 2008). In those works, a trajectory is already "there", but the cost is attributed to the act of observing a time point. In contrast, in our setting, we cannot directly acquire a time point, because there is a cost in the simulation of the trajectory.

**Multi-fidelity AL.** Closely related to our work is multi-fidelity active learning (S. Li et al., 2022b; Dongxia Wu et al., 2023; Hernandez-Garcia et al., 2023; S. Li et al., 2024), where outputs are acquired at varying fidelities for each input, with associated costs inherent to each fidelity. In our context, the task of actively selecting a sampling pattern for a given initial condition can be seen as a fidelity selection problem, where acquiring all time steps corresponds to the highest fidelity but also incurs the highest cost.

## 5. Experiments

### 5.1. Baseline AL Methods

To compare with our method, we experiment with AL for full trajectory sampling introduced in Musekamp et al. (2024). **Random** sampling from the pool set is the simplest method. **QbC** (Seung et al., 1992) is a simple active learning algorithm that selects points according to maximum disagreement among members of a committee. **LCMD** (Holzmüller et al., 2023) is an AL algorithm that uses a feature map. We concatenate the last hidden layer activations of committee members at all time steps of a trajectory, and sketch the concatenated features to a dimension of 512 using a random projection. Kirsch et al. (2023) proposes **SBAL**, which randomly samples data points $x$ with a probability distribution proportional to its temperature-scaled acquisition value $p(x) \propto a(x)^m$. We use the acquisition function of QbC with temperature $m = 1$. We

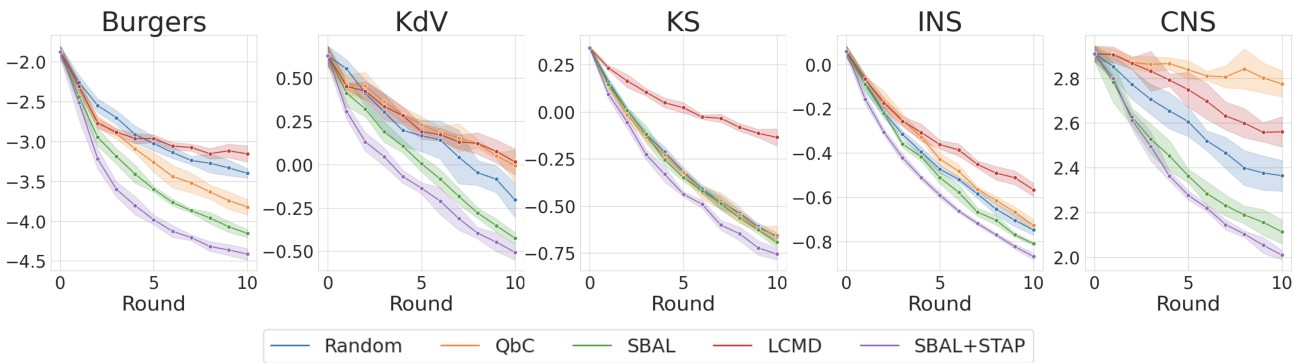

Figure 3: Log RMSE of AL strategies, measured across 10 rounds of acquisition. Each round incurs constant cost of data acquisition, namely the budget $B$.

leave out Core-Set (Sener and Savarese, 2017) because it generally underperforms compared to the above methods, according to both Holzmüller et al. (2023) and Musekamp et al. (2024).

### 5.2. Target PDEs

We evaluate our method on a range of PDEs. The first is the **Burgers'** equation, which shows either diffusive behavior or shock phenomena depending on the viscosity parameter. Next, we test the nonlinear Korteweg–De Vries (**KdV**) equation, which is known for exhibiting solitary wave pulses with weak interactions (Zabusky and Kruskal, 1965). We then apply our method to the Kuramoto–Sivashinsky (**KS**) equation, another nonlinear PDE in one dimension, notable for its chaotic dynamics. Lastly, we examine two forms of the Navier–Stokes equation in two spatial dimensions: the incompressible Navier–Stokes equation (**INS**) in vorticity form and the compressible Navier–Stokes equation (**CNS**). Both present significant challenges due to their strong nonlinearity and turbulent behavior. All equations are solved with periodic boundary conditions. Our choice of PDEs and their parameters is designed to validate our research across various flows exhibiting complex behaviors, including shock waves, chaotic dynamics, and turbulence. Additional details are in Appendix B.1.

### 5.3. Surrogate Models

We use Fourier Neural Operators (FNO, Z. Li et al., 2020) for our surrogate model $\hat{G}$. In particular, we train it to predict the differences between states in adjacent time steps, following Musekamp et al. (2024). All models have four hidden layers with 64 channels in each layer. We use 32, 256, 128, 16, and 32 Fourier modes for Burgers, KdV, KS, INS and CNS equations, respectively. We also normalize the data according to the initial dataset's mean and standard deviation over all temporal and spatial dimensions. The

Table 1: Log RMSE of AL strategies, averaged across 10 rounds of acquisition

| | Burgers | KdV | KS | INS | CNS |
|---|---|---|---|---|---|
| Random | -2.881±0.060 | 0.191±0.058 | -0.258±0.003 | -0.422±0.010 | 2.603±0.038 |
| QbC | -3.121±0.065 | 0.266±0.027 | -0.268±0.003 | -0.385±0.014 | 2.844±0.021 |
| LCMD | -2.847±0.027 | 0.256±0.030 | 0.046±0.013 | -0.320±0.011 | 2.736±0.029 |
| SBAL | -3.388±0.052 | 0.030±0.029 | -0.275±0.014 | -0.461±0.012 | 2.422±0.045 |
| SBAL+STAP | **-3.674**±0.071 | **-0.088**±0.040 | **-0.349**±0.003 | **-0.525**±0.005 | **2.363**±0.018 |

FNO is simply trained on ground truth input-output pairs from the solver $G$ without backpropagating through two or more time steps. All models were trained with Adam (Kingma, 2014) for 100 epochs, using a learning rate of $10^{-3}$, a batch size of 32, and a cosine annealing scheduler (Loshchilov and Hutter, 2016).

### 5.4. Results

We compare between the four baselines introduced in § 5.1, and our method combined with SBAL (SBAL+STAP). The pool set has 10,000 initial conditions, and we always start with an initial dataset of 32 fully sampled trajectories. The initial conditions in the test set are sampled from the same distribution as those in the pool set. An ensemble size of $M = 2$ is used, as it has been shown to be sufficient for good AL performance (Pickering et al., 2022; Musekamp et al., 2024). We perform 10 rounds of acquisition, and the budget of each round is set to $B = 8 \times L$ where $L$ is the length of a full trajectory. This means that at every round of AL, full trajectory algorithms sample 8 trajectories, each of length $L$. All experiments were conducted on 8 NVIDIA GeForce RTX 2080 Ti GPUs, and the results are averages from 5 seed values. Appendix C provides full report of all metrics on all methods, while here we summarize the most interesting results.

Fig. 3 shows plots of the committee's logarithmic RMSE across the 10 rounds of acquisition, and Table 1 summarizes the results with mean logarithmic RMSEs, where the

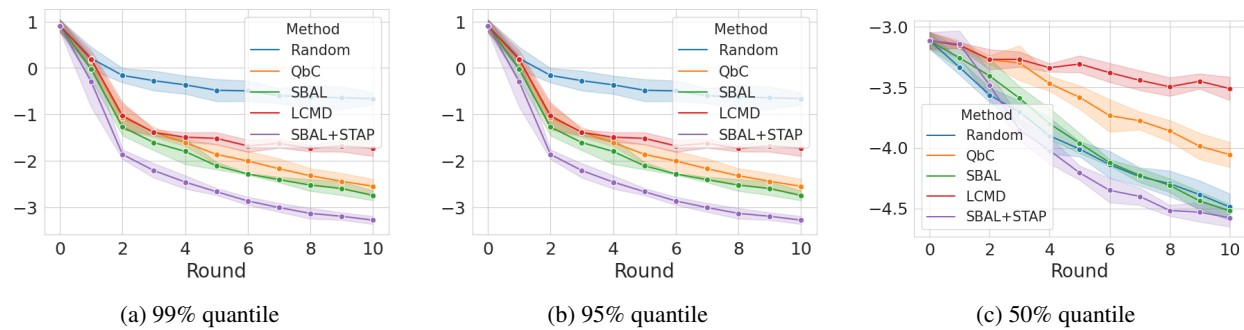

| (a) 99% quantile | (b) 95% quantile | (c) 50% quantile |
|---|---|---|

Figure 4: Quantiles of log RMSE on Burgers measured across 10 rounds of acquisition.

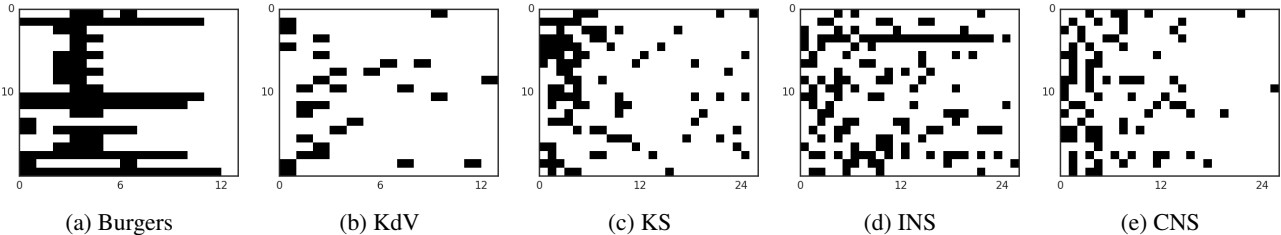

| (a) Burgers | (b) KdV | (c) KS | (d) INS | (e) CNS |
|---|---|---|---|---|

Figure 5: Timesteps chosen by SBAL+STAP. Each row corresponds to an acquired trajectory, where the black cells indicate the selected time steps. We show twenty trajectories acquired in the first rounds of active learning.

mean is taken over all 10 rounds. We can observe from the plots that SBAL+STAP outperforms other AL baselines in a robust manner. Most notably, it improves performance on the KS equation, where no other baseline improves significantly over Random selection.

Following Holzmüller et al. (2023), we also report the 99%, 95%, and 50% quantiles of log RMSE. Quantiles are useful for analyzing the behavior of AL strategies. AL methods tend to improve performance on points with extreme errors, or in other words the top quantiles, while not so much in the middle quantiles. This is why AL methods perform differently depending on the algorithm and the nature of the task. Fig. 4 shows their respective plots for Burgers. All baseline methods improve performance over random sampling in the 99% and 95% quantiles, and this is true even for LCMD which doesn't improve over random sampling in average error. In the 50% quantile, none of the baseline methods reduce the error below random sampling, but STAP surprisingly outperforms random sampling. This trend holds for all equations, as detailed in Appendix C. We can therefore infer that STAP isn't simply sacrificing the surrogate model's performance in some trajectories to improve its performance in others.

### 5.5. Random Bernoulli Sampling of Time Steps

We plot in Fig. 5 the distribution of time steps that STAP chooses. The plot clearly shows the general tendency of STAP to acquire the early time steps, with an occasional

Table 2: Log RMSE with Bernoulli sampling averaged across 10 rounds of acquisition.

|  | **Burgers** | **KdV** | **KS** | **INS** | **CNS** |
|---|---|---|---|---|---|
| SBAL | -3.388±0.052 | 0.030±0.029 | -0.275±0.014 | -0.461±0.012 | 2.422±0.045 |
| +STAP | **-3.674**±0.071 | **-0.088**±0.040 | -0.349±0.003 | -0.525±0.005 | **2.363**±0.018 |
| +Ber(1/16) | -3.231±0.163 | 0.053±0.014 | **-0.365**±0.008 | **-0.529**±0.006 | 2.375±0.069 |
| +Ber(1/8) | -3.152±0.267 | 0.049±0.014 | -0.359±0.006 | -0.525±0.006 | 2.370±0.018 |
| +Ber(1/4) | -3.102±0.441 | 0.018±0.024 | -0.346±0.008 | -0.515±0.007 | 2.372±0.012 |
| +Ber(1/2) | -3.458±0.067 | -0.064±0.031 | -0.324±0.007 | -0.500±0.009 | 2.392±0.012 |

selection of the later time steps. The distributions still show clear differences between tasks, such as the number of chosen time steps per trajectory or the frequency of later time steps. This suggests that STAP is choosing time steps in a highly adaptive manner.

We then ask ourselves: what if we perform random selection, for instance with a probability $p$, for every time step? We call this method Bernoulli sampling, or Ber($p$), where each entry of $S$ is true with probability $p$. Table 2 summarizes the performance of Ber($p$) for $p = 1/16, 1/8, 1/4$, and $1/2$. In general, Bernoulli sampling improves over the base method SBAL, but it can also severely underperform at certain values of $p$, such as for KdV. Still, for each PDE, there exists at least one value of $p$ at which Bernoulli sampling provides an advantage over the base method SBAL. In other words, sparse sampling of time steps itself has an inherent advantage over full-trajectory sampling, as first hypothesized with our analysis in Fig. 1.

Table 3: Wall-clock time of each procedure during batch selection in INS. Note that these are not the costs of data acquisition, but the computational cost of batch selection algorithms.

|  | Random | QbC | LCMD | SBAL |
|---|---|---|---|---|
| Time taken (seconds) | 0.1±0.1 | 62.5±0.1 | 100.3±0.1 | 62.5±0.1 |
|  | +STAP | +STAP MF | +STAP 10 |  |
| Time taken (seconds) | 132.7±0.7 | 91.5±0.1 | 15.4±0.1 |  |

Table 4: Log RMSE of more efficient STAP variants averaged across 10 rounds.

|  | SBAL | +STAP | +STAP MF | +STAP 10 |
|---|---|---|---|---|
| **Burgers** | -3.388±0.052 | **-3.674**±0.071 | -3.608±0.177 | -3.524±0.082 |
| **KdV** | 0.030±0.029 | **-0.088**±0.040 | -0.065±0.034 | -0.118±0.024 |
| **KS** | -0.275±0.014 | **-0.349**±0.003 | -0.326±0.004 | -0.316±0.009 |
| **INS** | -0.422±0.010 | -0.525±0.005 | **-0.529**±0.005 | -0.502±0.004 |
| **CNS** | 2.422±0.045 | 2.363±0.016 | **2.360**±0.005 | 2.401±0.004 |

How does STAP manage to be on par with or outperform the best $\mathrm{Ber}(p)$ on every task? One might hypothesize that the frequency of chosen time steps is what matters most for performance and that STAP somehow finds the optimal frequency $p$ to sample with. We have measured the frequency with which STAP samples time steps in the first round of AL, and obtained 0.35, 0.11, 0.19, 0.22, and 0.16 for Burgers, KdV, KS, INS, and CNS, respectively. One can see that for KdV, despite $\mathrm{Ber}(p)$ performing better with higher $p$ and best with $p = 1/2$, STAP outperforms all $\mathrm{Ber}(p)$ with a frequency of 0.11, which is close to 1/8. In other words, the specific time steps that are sampled also matter as much as the overall frequency. These observations altogether suggest that STAP adaptively chooses not only the frequency of the time steps to acquire, but also their specific locations. We report the full results in Appendix D.3, along with a variant of Bernoulli sampling that enforces acquiring consecutive initial time steps.

### 5.6. Computational Complexity of STAP

The time complexity of computing our acquisition function for a single instance of $(\boldsymbol{u}^0, S)$ is $O(M^2 L)$. Since we optimize the acquisition function with $T$ steps, and we can acquire at most $B$ initial conditions, the time complexity of our batch acquisition algorithm is $O(M^2 LBT)$ in the worst case. We can parallelize the optimization of multiple $S_j$'s to a certain extent using graphics processing unit (GPU), which can significantly alleviate the burden of $B$.

We can further reduce the cost by at most a factor of $M$ with STAP MF described in Appendix B.4. Yet another al-

Table 5: Log RMSE with multistep FNO averaged across 10 rounds of acquisition.

|  | Random | SBAL | +STAP |
|---|---|---|---|
| **Burgers 8L/8** | -1.670±0.098 | -1.893±0.053 | **-2.058**±0.028 |
| **KdV 8L/8** | 1.402±0.029 | 1.404±0.024 | **1.364**±0.043 |
| **KS 8L/8** | 1.255±0.015 | 1.232±0.012 | **1.156**±0.008 |
| **KS L/2** | 1.340±0.014 | 1.335±0.011 | **1.288**±0.011 |
| **INS L/2** | 1.124±0.017 | 1.118±0.007 | **1.081**±0.012 |
| **CNS L/2** | 3.593±0.023 | 3.594±0.044 | **3.420**±0.050 |

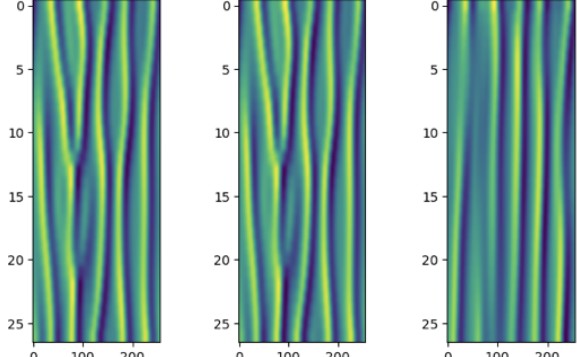

Figure 6: From left to right, ground truth trajectory on KS, trajectory predicted by surrogate model trained with 32 trajectories, and trajectory predicted by surrogate model trained with one trajectory.

ternative is to decrease the number of greedy optimization steps $T$ from 100 to 10, which reduces the cost by a factor of 10. We call this variant STAP 10. The wall-clock time of each baseline method and STAP is measured with a single NVIDIA GeForce RTX 2080 Ti GPU, and summarized in Table 3. The performance of SBAL with STAP and its two variants are summarized in Table 4. Note that STAP 10 incurs only a fraction of computational cost over the baseline methods, while still achieving a significant boost in performance over its base method.

### 5.7. Multi-step model

We have done experiments with multi-step FNO which receives N timesteps as input and outputs N timesteps. To perform STAP, we group the total number of timesteps into non-overlapping clusters of N timesteps, and perform STAP as if each cluster is one timestep with N channels. We have performed two variants of experiments. In the first, we divide a timestep in our main experiment into 8 smaller timesteps, so that a total of $L$ timesteps turn into $8L$ timesteps, and train 8-in-8-out models. In the second variant, we keep $L$ timesteps the same but train 2-in-2-out models. Fig. 13 and Table 5 shows the log RMSE for Burg-

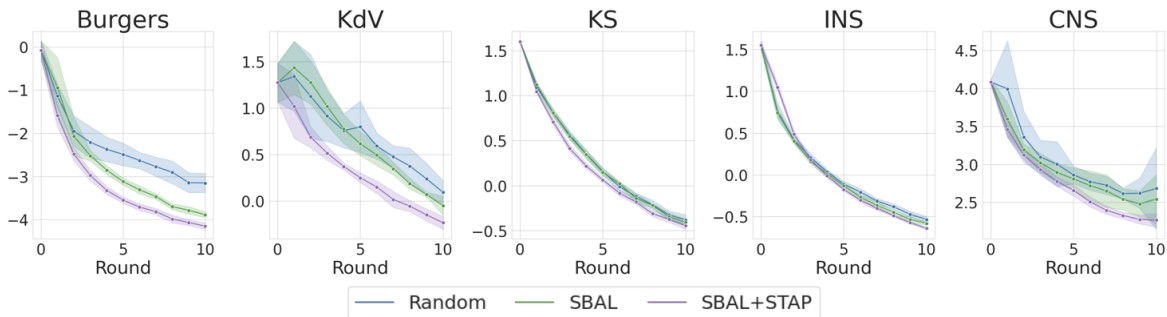

Figure 7: Active learning with initial training dataset containing one trajectory, as opposed to 32 trajectories in the main experiment. STAP is robust to initially inaccurate surrogate models.

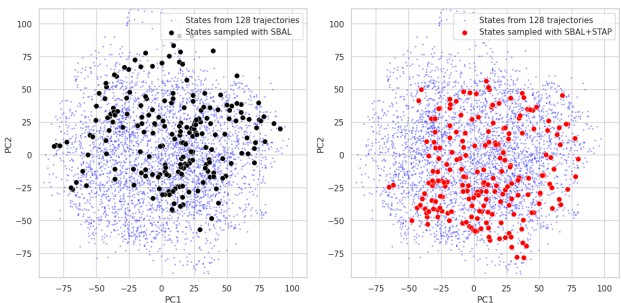

Figure 8: PCA of FNO hidden layer activation for PDE states sampled by SBAL (left) and SBAL+STAP (right) during the first round of active learning on KS.

ers, KdV, and KS of the first variant, and KS, INS and CNS of the second variant.

### 5.8. Out-of-distribution Synthetic Inputs

Inaccurate surrogate models might synthesize inputs that lie far from the ground truth distribution, harming the representativeness (Dongrui Wu, 2018). Indeed, under limited training data, the surrogate model outputs visibly erroneous trajectories. Fig. 6 are comparisons of the ground truth trajectories and predictions from surrogate models trained on 32 trajectories and one trajectory, respectively.

To test how much this error harms STAP, we perform experiments where the initial training dataset contains 1 trajectory, compared to 32 used in our main experiments. Fig. 7 shows the log RMSEs. To our own surprise, we find that SBAL+STAP still outperforms Random and SBAL, except for INS in the early rounds.

Fig. 8 shows comparisons of FNO activation PCAs for PDE states sampled by SBAL and SBAL+STAP during the first round of active learning on KS. Note that the PCA was fitted to ground truth states in random trajectories (blue points), independent from the states sampled by SBAL and SBAL+STAP, so that out-of-distributionness

can be properly evaluated. We find that only several of the states sampled by SBAL+STAP diverge significantly from the random ground truth states. In other words, the surrogate model synthesizes erroneous inputs when looked at trajectory-wise, but when looked at individually, they aren't out-of-distribution enough to harm STAP significantly.

## 6. Conclusion

In this paper, we presented a novel framework for active learning in surrogate modeling of partial differential equation (PDE) trajectories, significantly reducing the cost of data acquisition while maintaining or improving model accuracy. By selectively querying only a subset of time steps in a PDE trajectory, our method STAP enables the acquisition of informative data at a fraction of the cost of acquiring entire trajectories. We introduced a new acquisition function that estimates the utility of a set of time steps based on variance reduction, effectively guiding the selection process in an adaptive manner. Through extensive experiments on benchmark PDEs, including the Burgers equation, Korteweg–De Vries equation, Kuramoto–Sivashinsky equation, incompressible Navier-Stokes equation, and compressible Navier-Stokes equation, we demonstrated that our approach consistently outperforms existing AL methods, providing a more cost-efficient and accurate solution for PDE surrogate modeling.

Our results show that STAP can significantly enhance surrogate modeling for PDEs, particularly in scenarios where the numerical solver is computationally expensive. We further showed that the success of STAP is driven by its ability to prioritize both diverse and informative time steps. Moving forward, this framework could be extended to more complex systems and integrated with other machine learning techniques, providing broader applicability in scientific and engineering simulations. Future work may also explore alternative acquisition functions and applications to simulations outside the domain of PDEs.

## Acknowledgements

This work was partly supported by Institute of Information & communications Technology Planning & Evaluation(IITP) grant funded by the Korea government(MSIT) (No.RS-2024-00509279, Global AI Frontier Lab; No.RS-2019-II190075, Artificial Intelligence Graduate School Program(KAIST); No.RS-2022-II220184, Development and Study of AI Technologies to Inexpensively Conform to Evolving Policy on Ethics), and the National Research Foundation of Korea(NRF) grant funded by the Korea government(MSIT) (NRF-2022R1A5A708390812). We also thank Hyungi Lee for his thoughtful discussion.

## Impact Statement

We propose a new method that improves the cost efficiency of acquiring data for building a surrogate model of PDE trajectories. Although our approach doesn't have a direct positive or negative impact in ethical or societal aspects, it accelerates the process of building a surrogate model for an arbitrary PDE. This could be used for good, such as medical simulations, environmental modeling, and optimizing engineering designs, potentially leading to advancements in healthcare, sustainability, and technological innovation. However, like many technologies, this method could also be misused in domains where rapid simulations could have harmful consequences, such as the development of hazardous materials. Therefore, researchers and practitioners should apply these methods with consideration of their broader societal implications, aiming to ensure that the benefits of the technology are used responsibly and ethically.

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

# A. Further explanation of Acquisition with STAP

## A.1. Different types of PDE active learning

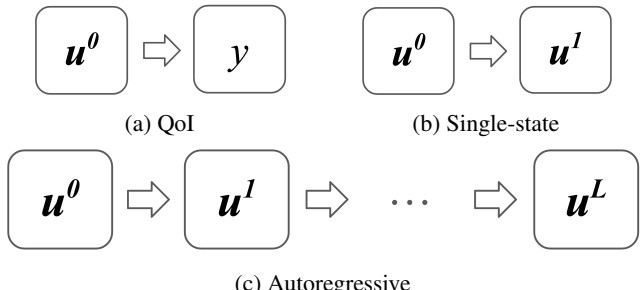

Figure 10: Task settings assumed by previous works in active learning of PDEs.

There are three primary tasks in active learning for PDEs, each depending on the type of surrogate model being trained. The first task, *univariate Quantity of Interest (QoI) prediction*, focuses on learning a model to directly predict a scalar QoI, denoted as $y$, from an initial condition $\boldsymbol{u}^0$. The second task, *single-state prediction*, involves learning a model to predict a single state transition from $\boldsymbol{u}^0$ to $\boldsymbol{u}^1$ over a fixed time interval $\Delta t$. The third task, *autoregressive trajectory prediction*, aims to approximate the ground truth evolution operator $G$ using a surrogate model to predict the entire time evolution of the states. Fig. 10 provides a visual comparison of the three tasks. In this paper, we focus on the autoregressive trajectory prediction task.

## A.2. Motivation behind the Acquisition Function

Here we detail the motivation behind our acquisition function defined in § 3.2. First, one can imagine several alternative acquisition functions.

The most straightforward alternative is to simply use the sum of the variances at time points for which $b_i = $ true. The variances are larger for the later time steps since they accumulate, and in our preliminary experiments, we found that this is catastrophic as undersampling the earlier time steps leads to the sampled trajectory being very out-of-distribution, and hence the trained surrogate model underperforming on the test distribution.

It quickly became clear to us that we need some kind of measure of "how much total uncertainty will be reduced by sampling these time steps", instead of "how uncertain is our model on these time steps?" This would help select sampling patterns that reduce the out-of-distribution-ness introduced by $\hat{G}$. One way to approximate this is to use mutual information, as used by S. Li et al. (2022b). In other words, we would rollout $M$ trajectories with $M$ surrogate models, and compute the mutual information between time steps for which $b_i = $ true and all time steps. However, in preliminary experiments, we found that this method underperforms, which we hypothesize is because relying simply on the covariance matrix of the committee between time steps is not a good enough method for computing the posterior uncertainty.

We identified two "pathways" through which sampling a time step reduces uncertainty in the remaining time steps. First, there is the "indirect" pathway: sampling a time step will reduce the model's uncertainty on similar inputs, hence reducing uncertainty on the remaining time steps. This is what is approximated by mutual information. Then, there is the "direct" pathway: sampling a time step $i$ gives out the $i + 1$ th state, which starts a chain reaction of reducing model uncertainty on all successive states. Note that these two pathways are not distinct from a strictly theoretical view, but are rather two ways of approximating uncertainty reduction.

The direct pathway motivated our acquisition function based on variance reduction. In variance reduction, we calculate the posterior uncertainty by rolling out the trajectories with $N$ surrogate models, but collapse into one surrogate model at time steps for which $b_i = $ true. This effectively computes the reduced uncertainty due to the effect of the direct pathway. With experiments, we confirmed that this acquisition function behaves just like we wanted: it is slightly biased towards sampling the earlier time steps, and it chooses an appropriate frequency of time steps to sample that leads to good performance.

To elaborate, our acquisition function is an approximation to the expected error reduction (EER), which is statistically near-optimal for active learning (Settles, 2009; Roy and McCallum, 2001). The EER measures how much the model's gen-

---

**Algorithm 2** Batch Acquisition Algorithm of STAP

---

**Require:** Budget $B$, base active learning algorithm $\mathcal{A}$, probability $\epsilon$, number of iterations $T$ for greedy optimization, pool $P$ of initial conditions, cost function $\text{cost}(\cdot)$ for batches.
**Ensure:** A batch $\mathcal{B}$ of initial conditions and sampling patterns.
 1: $\mathcal{B} \leftarrow \varnothing$
 2: **while** $\text{cost}(\mathcal{B}) < B$ **do**
 3:     Acquire an initial condition $\boldsymbol{u}^0$ with $\mathcal{A}$.
 4:     Initialize $S \leftarrow (\text{true}, \dots, \text{true})$.
 5:     **for** $i = 1$ to $T$ **do**
 6:         $C = (C_1, \dots, C_L)$ where $C_1, \dots, C_L \overset{\text{i.i.d.}}{\sim} \text{Ber}(\varepsilon)$.
 7:         $S' = S \oplus C$
 8:         **if** $a^*(\boldsymbol{u}^0, S') \geq a^*(\boldsymbol{u}^0, S)$ **then**
 9:             $S \leftarrow S'$.
10:         **end if**
11:     **end for**
12:     **if** $\|S\| + \text{cost}(\mathcal{B}) > B$ **then**
13:         Keep only the first $(B - \text{cost}(\mathcal{B}))$ trues from $S$ and flip the remaining trues.
14:     **end if**
15:     $\mathcal{B} \leftarrow \mathcal{B} \cup \{(\boldsymbol{u}^0, S)\}$.
16: **end while**

---

eralization error is likely reduced after updating on hypothetically acquired data. We model our hypothetical belief about the ground truth solver as a *uniform categorical distribution* over the ensemble $\{\hat{G}_a\}_{a=1}^M$. We assume that acquiring the trajectory of $u^0$ with sampling pattern $S$ only reduces generalization error on the trajectory of $u^0$. The current generalization error is expected to be the average of $\|\hat{u}_a - \hat{u}_b\|^2$ over $b$. We make a second assumption that the hypothetically acquired data $\hat{u}_{b,S,a}$ will update the model such that the model predicts the trajectory $\hat{u}_{b,S,a}$ given $u^0$. This gives us the expected reduction in error $\|\hat{u}_a - \hat{u}_b\|^2 - \|\hat{u}_a - \hat{u}_{b,S,a}\|^2$ averaged across $a$ and $b$, which is equal to our acquisition function.

### A.3. Batch Acquisition Algorithm

Algorithm 2 summarizes the batch selection algorithm of STAP. Starting with an empty batch $\mathcal{B}$, the algorithm repeatedly selects initial conditions and their sampling patterns until reaching the budget limit. It first uses the base active learning method $\mathcal{A}$ to choose an initial condition $\boldsymbol{u}^0$. Then, it optimizes which time steps to sample through a greedy procedure: starting with a pattern $S$ that samples all time steps (all true values), it performs $T$ iterations of random mutations. In each iteration, it generates a candidate pattern $S'$ by randomly flipping entries in $S$ with probability $\epsilon$ (using a binary mask $C$ where each entry is drawn from a Bernoulli distribution and the XOR operation $\oplus$). If this new pattern achieves a better value according to the cost-weighted acquisition function $a^*$, it becomes the current pattern. To ensure the budget isn't exceeded, if adding the current pattern would go over budget, the algorithm truncates it by keeping only enough true values to exactly meet the budget. The pair of initial condition and its optimized sampling pattern $(\boldsymbol{u}^0, S)$ is then added to the batch $\mathcal{B}$.

## B. Experimental details

### B.1. Details on PDEs

In this section, we describe the PDEs used in our experiments. Each of these equations plays a critical role in modeling physical phenomena and showcases diverse behaviors, from diffusion and soliton dynamics to chaotic systems and fluid flow. Examples of PDE trajectories are shown in Fig. 11.

**Burgers' Equation**   The one-dimensional Burgers' equation is expressed as

$$\partial_t u + u \partial_x u = (\nu/\pi) \partial_{xx} u \tag{7}$$

where $u = u(x,t)$ represents the field and $\nu \geq 0$ is the viscosity parameter. The Burgers' equation exhibits shock phenomena, characterized by the gradients of $u$ becoming extremely sharp or even discontinuous when $\nu = 0$. These

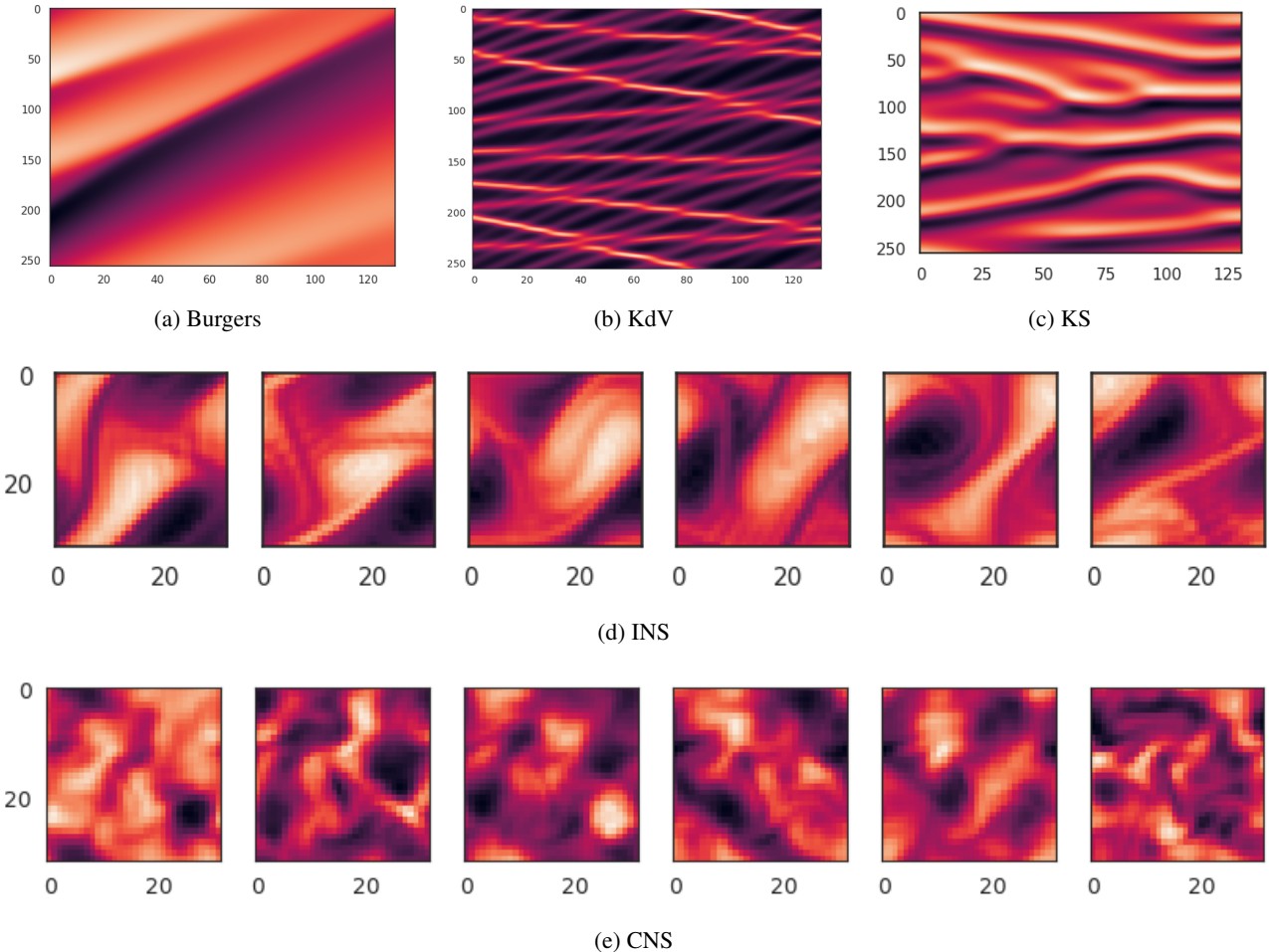

(a) Burgers         (b) KdV         (c) KS

(d) INS

(e) CNS

Figure 11: Example trajectories of PDEs. (a), (b), (c): Horizontal and vertical axes represent the temporal and spatial domain. (d), (e): Two-dimensional states at six time points are shown.

shocks arise due to the the advection term $u\partial_x u$, while the presence of the diffusion term $\partial_{xx} u$ prevents the formation of discontinuities in the wave. We set $\nu = 0.01$ to ensure the formation of sharp gradients while maintaining numerical stability. For simulations, we adopt the implementation by Takamoto et al. (2022), which employs a finite difference method (FDM) to compute the two terms.

**Korteweg–De Vries (KdV) Equation** The second equation we study is the Korteweg–De Vries (KdV) equation, given by:

$$\partial_t u + u\partial_x u + \partial_{xxx} u = 0, \tag{8}$$

where $u = u(x,t)$ represents a wave profile evolving over space and time. This nonlinear PDE describes the evolution of shallow water waves, and its most famous characteristic is the presence of solitons—solitary, stable wave packets that maintain their shape over long distances and weak interactions with other waves (Zabusky and Kruskal, 1965). Solitons have important applications in fluid dynamics, plasma physics, and optical fiber communications. The KdV equation's nonlinearity and third-order spatial derivative ($\partial_{xxx}$) allow it to capture complex wave behavior. The equation is also known for conserving key quantities like energy. We solve this equation using the pseudospectral method with Dormand–Prince solver as in Brandstetter et al. (2022a).

**Kuramoto–Sivashinsky (KS) Equation** The Kuramoto–Sivashinsky (KS) equation is a fourth-order nonlinear PDE, written as:

$$\partial_t u + \partial_{xx} u + \partial_{xxxx} u + u\partial_x u = 0, \tag{9}$$

where $u = u(x, t)$ is the evolving field in space and time. The KS equation is known for its chaotic behavior and is used to model phenomena such as flame front propagation, plasma instabilities, and thin film dynamics. Its chaotic nature arises from the interplay between destabilizing nonlinear terms and stabilizing higher-order diffusion terms. The equation is particularly challenging to solve due to its sensitivity to initial conditions and long-term unpredictability. To handle this complexity, we use the Exponential Time Differencing (ETD) fourth-order Runge-Kutta method, as introduced by Kassam and Trefethen (2005). This numerical method is well-suited for stiff PDEs like the KS equation.

**Incompressible Navier-Stokes (INS) Equation**    We consider the vorticity form of the 2D incompressible Navier-Stokes (INS) equation, which governs the motion of incompressible and viscous fluid flows. The equation is given by:

$$\partial_t u + \boldsymbol{v} \cdot \nabla u = \nu \nabla^2 u + f, \quad \nabla \cdot \boldsymbol{v} = 0, \tag{10}$$

where $u(x_1, x_2, t)$ is the vorticity, $\boldsymbol{v}$ is the velocity field, $\nu$ is the kinematic viscosity, and $f(x_1, x_2)$ is an external forcing term. This equation describe the behavior of incompressible fluid flow, playing a central role in understanding turbulence, weather patterns, and aerodynamics. The external forcing term $f(x_1, x_2)$ is set to

$$f(x) = 0.1 \left( \sin(2\pi(x_1 + x_2)) + \cos(2\pi(x_1 + x_2)) \right), \tag{11}$$

which injects energy into the system, driving complex fluid dynamics. We use kinematic viscosity of $\nu = 2 \times 10^{-4}$, ensuring a sufficiently turbulent regime. In our experiments, we adapt the Crank–Nicolson method implemented by Z. Li et al. (2020).

**Compressible Navier-Stokes (CNS) Equation**    Finally, we consider the 2D compressible Navier-Stokes (CNS) equation, which extends the incompressible case by incorporating density variations. We adapt the form used in Takamoto et al. (2022), which is given by:

$$\partial_t \rho + \nabla \cdot (\rho \boldsymbol{v}) = 0,$$
$$\rho(\partial_t \boldsymbol{v} + \boldsymbol{v} \cdot \nabla \boldsymbol{v}) = -\nabla p + \eta \triangle \boldsymbol{v} + (\zeta + \frac{1}{3}\eta)\nabla(\nabla \cdot \boldsymbol{v}), \tag{12}$$
$$\partial_t(\epsilon + \frac{1}{2}\rho \boldsymbol{v}^2) + \nabla \cdot [(p + \epsilon + \frac{1}{2}\rho \boldsymbol{v}^2)\boldsymbol{v} - \boldsymbol{v} \cdot \sigma'] = 0,$$

where it has four variables (density $\rho$, velocity $\boldsymbol{v} = (v_x, v_y)$ and pressure $p$) and two fixed parameters (shear viscosity $\eta$ and bulk viscosity $\zeta$). The terms $\sigma'$ and $\epsilon$ are the viscous stress tensor and the internal energy, respectively. We set $\eta = \zeta = 0.05$. The numerical solution was computed using a second-order scheme in both time and space, employing a high-resolution reconstruction method for the inviscid terms and a central differencing approach for the viscous terms.

**Initial conditions**    As per Brandstetter et al. (2022a), states are first sampled from a simple distribution and then evolved for a certain time to obtain the initial conditions. The evolved initial conditions are more realistic than the sampled states, in that they are more likely to be observed under a system governed by the respective PDEs. This procedure hence approximates applications where the initial conditions of interest are realistic states either from observed data (Jumper et al., 2021; Kalnay, 2003; Chassignet et al., 2007; Taylor et al., 2012) or carefully crafted synthetic data (Jarrin et al., 2006; Kusner et al., 2017). For 1D equations, the states are sampled from truncated Fourier series with random coefficients (Brandstetter et al., 2022a), and for the 2D INS equation, states are sampled from a Gaussian random field as described in Z. Li et al. (2020). Similarly, for the CNS equation, the initial conditions for all four variables are constructed by superposing random sinusoidal waves, with density and pressure renormalized to ensure positivity. The lengths and discretizations of trajectories are summarized in Table 6.

Let $U$ stand for the uniform distribution. For Burgers and KdV, the initial condition is in the form $\sum_{i=1}^{N} A_i \sin(2\pi k_i x / L + \phi_i)$. The amplitudes and phases are always sampled from $U([0, 1])$ and $U([0, 2\pi])$. For Burgers, $N = 2$ and $k_i \sim U(1, 2, 3, 4)$, and for KdV, $N = 10$ and $k_i \sim U(1, 2, 3)$. For KS and INS, the initial conditions are gaussian random fields drawn from $N(0, 25(-\Delta + 25I)^{-1})$ and $N(0, 7^{3/2}(-\Delta + 49I)^{-2.5})$ respectively. For CNS, we first sample $\sum_{k \in \{1,2,3,4\}^3} A_k \sin(2\pi k x / L + \phi_k)$, the amplitudes and phases sampled uniformly at random for each channel ($rho$, $p$ and $v$). Then, we renormalize $rho$, $p$ to lie within $\rho_0(1 \pm \Delta_\rho)$ and $p_0(1 \pm \Delta_p)$, respectively, where $\rho_0 \sim U([0.1, 10])$, $\Delta_\rho \sim U([0.013, 0.26])$, $\Delta_p \sim U([0.04, 0.8])$ and $T_0 := \rho_0/p_0 \sim U([0.1, 10])$. The $v$ is also computed by superposing sinusoidal waves, but with amplitudes chosen so that the initial condition has the given initial Mach number $M \sim U([0.1, 1])$.

Table 6: Domain lengths and discretizations for trajectory learning.

| PDE | Domain Length $(T, X)$ | Resolution $(L, N_x)$ |
|-----|------------------------|------------------------|
| Burgers | (2.0, 1.0) | (13, 256) |
| KdV | (52.0, 128.0) | (13, 256) |
| KS | (13.0, 1.0) | (26, 256) |
| INS | (13.0, 1.0, 1.0) | (26, 32, 32) |
| CNS | (0.5, 1.0, 1.0) | (26, 32, 32) |

Table 7: Acquired datasize in KdV

| Round | 0 | 1 | 2 | 3 | 4 | 5 | 6 | 7 | 8 | 9 | 10 |
|-------|-----|-----|-----|-----|-----|-----|------|------|------|------|------|
| SBAL | 416 | 520 | 624 | 728 | 832 | 936 | 1040 | 1144 | 1248 | 1352 | 1456 |
| SBAL+STAP | 416 | 507 | 611 | 715 | 819 | 923 | 1027 | 1131 | 1235 | 1339 | 1443 |

## B.2. Error Metrics

The test set always consists of 1,000 trajectories, on which several error metrics are defined. The **RMSE** is defined on a trajectory $\boldsymbol{u}$ as

$$\sqrt{\frac{1}{LN_x} \sum_{i=1}^{L} \sum_{j=1}^{N_x} \|\boldsymbol{u}^i(\mathbf{x}_j) - \hat{\boldsymbol{u}}^i(\boldsymbol{x}_j)\|_2^2}. \tag{13}$$

Similarly, the **NRMSE** is defined as

$$\sqrt{\frac{\sum_{i,j} \|\boldsymbol{u}^i(\mathbf{x}_j) - \hat{\boldsymbol{u}}^i(\boldsymbol{x}_j)\|_2^2}{\sum_{i,j} \|\boldsymbol{u}^i(\mathbf{x}_j)\|_2^2}} \tag{14}$$

and the **MAE** as

$$\frac{1}{LN_x} \sum_{i=1}^{L} \sum_{j=1}^{N_x} |\boldsymbol{u}^i(\mathbf{x}_j) - \hat{\boldsymbol{u}}^i(\boldsymbol{x}_j)|. \tag{15}$$

The metrics are averaged across all trajectories in the test set. We also report their logarithmic values averaged across all AL rounds, following Holzmüller et al. (2023). Note that we do not use a committee's mean prediction for computing the metrics, but instead compute the metrics for each model and report their average.

## B.3. Simulation Instability

It was observed that using STAP on the KdV equation, the simulation crashes on a small subset of synthetic inputs. Analysis reveals that these synthetic inputs have unusually large norms and particularly appear in later parts of trajectories due to accumulated error. We do not attempt to fix this problem explicitly due to the risk of over-complicating our method, and simply refrain from adding these time steps to the training dataset. This means that STAP actually acquires a smaller number of time steps than the budget $B$ per round of acquisition, which could be problematic when a large subset of inputs do crash. However, we find that this is not the case, and the number of such inputs is small enough that STAP can outperform other baselines. We report the comparison of datasize across rounds in Table 7, for a single experiment. We can see that 13 time steps were left out in the first round due to instability, and no instability occurred in the rounds after.

Since queries that crash incur a cost, they should be avoided as much as possible. Previous works in Bayesian optimization (Gelbart et al., 2014; Hernández-Lobato et al., 2015) propose methods to learn these unknown constraints. Alternatively, one could simply test out large, random inputs. In fact, we find that the maximum absolute value of an input being above 10 is a robust criterion for predicting that the solver will crash. Either way, we could simply filter out time steps that fall outside of these constraints during runtime of the solver, and use the freed up budget on acquiring other trajectories. Another possible approach is to impose physical constraints on the surrogate model (Goswami et al., 2022) that reduces the risk of outputting abnormal synthetic inputs. For instance, the KdV equation is energy-conserving, and when this prior knowledge is encoded into the surrogate model, the synthetic inputs would never be abnormally large like we experienced with our naive surrogate models.

**B.4. STAP MF**

We can also define a simpler acquisition function in the spirit of mean-field approximation. We take the mean model $\hat{G} = \frac{1}{M}\sum_m \hat{G}_m$, and define the variance reduction $R(\hat{G}, b, S)$ between $\hat{G}$ and a model $\hat{G}_b$ in the same way as before. We then average the variance reduction between the mean model and all models in the committee:

$$a_{\text{MF}}(\boldsymbol{u}^0, S) = \frac{1}{M}\sum_{b=1}^{M} R(\hat{G}, b, S), \tag{16}$$

which reduces the computational cost by a factor of $M$ in the best case. We call this modified version STAP MF.

## C. Full Report of Results on Main Experiment

Table 8: Mean log metrics for Burgers' Equation

|  | RMSE | NRMSE | MAE | 99% | 95% | 50% |
|---|---|---|---|---|---|---|
| Random | $-2.881_{\pm 0.060}$ | $-3.955_{\pm 0.060}$ | $-4.813_{\pm 0.067}$ | $-0.288_{\pm 0.145}$ | $-1.701_{\pm 0.070}$ | $-3.924_{\pm 0.067}$ |
| Random+STAP | $-3.477_{\pm 0.064}$ | $-4.433_{\pm 0.078}$ | $-5.394_{\pm 0.061}$ | $-1.010_{\pm 0.142}$ | $-2.551_{\pm 0.109}$ | $-4.278_{\pm 0.093}$ |
| SBAL | $-3.388_{\pm 0.052}$ | $-4.081_{\pm 0.061}$ | $-5.332_{\pm 0.046}$ | $-1.677_{\pm 0.065}$ | $-2.375_{\pm 0.067}$ | $-3.885_{\pm 0.047}$ |
| SBAL+STAP | $-3.674_{\pm 0.071}$ | $-4.306_{\pm 0.065}$ | $-5.585_{\pm 0.066}$ | $-2.185_{\pm 0.085}$ | $-2.853_{\pm 0.078}$ | $-4.017_{\pm 0.070}$ |
| QbC | $-3.121_{\pm 0.065}$ | $-3.805_{\pm 0.062}$ | $-5.067_{\pm 0.058}$ | $-1.483_{\pm 0.081}$ | $-2.098_{\pm 0.078}$ | $-3.571_{\pm 0.068}$ |
| QbC+STAP | $-3.333_{\pm 0.064}$ | $-3.941_{\pm 0.069}$ | $-5.250_{\pm 0.056}$ | $-1.874_{\pm 0.084}$ | $-2.453_{\pm 0.065}$ | $-3.674_{\pm 0.068}$ |
| LCMD | $-2.847_{\pm 0.027}$ | $-3.555_{\pm 0.022}$ | $-4.819_{\pm 0.027}$ | $-1.160_{\pm 0.041}$ | $-1.758_{\pm 0.015}$ | $-3.338_{\pm 0.049}$ |
| LCMD+STAP | $-2.925_{\pm 0.061}$ | $-3.546_{\pm 0.061}$ | $-4.886_{\pm 0.059}$ | $-1.281_{\pm 0.085}$ | $-1.874_{\pm 0.072}$ | $-3.360_{\pm 0.079}$ |

Table 9: Mean log metrics for KdV Equation

|  | RMSE | NRMSE | MAE | 99% | 95% | 50% |
|---|---|---|---|---|---|---|
| Random | $0.191_{\pm 0.058}$ | $-1.193_{\pm 0.050}$ | $-2.034_{\pm 0.045}$ | $2.449_{\pm 0.047}$ | $1.395_{\pm 0.049}$ | $-1.196_{\pm 0.043}$ |
| Random+STAP | $-0.067_{\pm 0.054}$ | $-1.425_{\pm 0.047}$ | $-2.228_{\pm 0.036}$ | $1.885_{\pm 0.033}$ | $1.296_{\pm 0.038}$ | $-1.424_{\pm 0.044}$ |
| SBAL | $0.030_{\pm 0.029}$ | $-1.282_{\pm 0.030}$ | $-2.139_{\pm 0.027}$ | $1.875_{\pm 0.039}$ | $1.267_{\pm 0.028}$ | $-1.267_{\pm 0.027}$ |
| SBAL+STAP | $-0.088_{\pm 0.040}$ | $-1.378_{\pm 0.040}$ | $-2.239_{\pm 0.043}$ | $1.731_{\pm 0.040}$ | $1.280_{\pm 0.043}$ | $-1.378_{\pm 0.040}$ |
| QbC | $0.266_{\pm 0.027}$ | $-1.019_{\pm 0.029}$ | $-1.879_{\pm 0.029}$ | $1.859_{\pm 0.037}$ | $1.251_{\pm 0.029}$ | $-1.019_{\pm 0.031}$ |
| QbC+STAP | $0.134_{\pm 0.035}$ | $-1.130_{\pm 0.037}$ | $-2.004_{\pm 0.035}$ | $1.721_{\pm 0.031}$ | $1.120_{\pm 0.037}$ | $-1.286_{\pm 0.035}$ |
| LCMD | $0.256_{\pm 0.030}$ | $-1.033_{\pm 0.036}$ | $-1.879_{\pm 0.033}$ | $1.868_{\pm 0.034}$ | $1.322_{\pm 0.034}$ | $-1.100_{\pm 0.038}$ |
| LCMD+STAP | $0.286_{\pm 0.034}$ | $-0.978_{\pm 0.034}$ | $-1.824_{\pm 0.039}$ | $1.799_{\pm 0.036}$ | $1.128_{\pm 0.034}$ | $-1.129_{\pm 0.032}$ |

Table 10: Mean log metrics for KS Equation

|  | RMSE | NRMSE | MAE | 99% | 95% | 50% |
|---|---|---|---|---|---|---|
| Random | $-0.258_{\pm 0.004}$ | $-1.683_{\pm 0.004}$ | $-2.165_{\pm 0.004}$ | $1.097_{\pm 0.003}$ | $0.752_{\pm 0.005}$ | $-0.575_{\pm 0.004}$ |
| Random+STAP | $-0.335_{\pm 0.014}$ | $-1.759_{\pm 0.014}$ | $-2.248_{\pm 0.014}$ | $1.060_{\pm 0.015}$ | $0.691_{\pm 0.007}$ | $-0.662_{\pm 0.015}$ |
| SBAL | $-0.275_{\pm 0.014}$ | $-1.700_{\pm 0.014}$ | $-2.184_{\pm 0.014}$ | $1.086_{\pm 0.017}$ | $0.732_{\pm 0.023}$ | $-0.594_{\pm 0.012}$ |
| SBAL+STAP | $-0.349_{\pm 0.003}$ | $-1.774_{\pm 0.003}$ | $-2.265_{\pm 0.003}$ | $1.042_{\pm 0.011}$ | $0.672_{\pm 0.012}$ | $-0.674_{\pm 0.008}$ |
| QbC | $-0.268_{\pm 0.004}$ | $-1.693_{\pm 0.004}$ | $-2.178_{\pm 0.004}$ | $1.077_{\pm 0.008}$ | $0.739_{\pm 0.013}$ | $-0.582_{\pm 0.006}$ |
| QbC+STAP | $-0.331_{\pm 0.014}$ | $-1.756_{\pm 0.014}$ | $-2.246_{\pm 0.014}$ | $1.050_{\pm 0.013}$ | $0.681_{\pm 0.020}$ | $-0.650_{\pm 0.013}$ |
| LCMD | $0.046_{\pm 0.015}$ | $-1.378_{\pm 0.015}$ | $-1.829_{\pm 0.015}$ | $1.204_{\pm 0.009}$ | $0.954_{\pm 0.011}$ | $-0.203_{\pm 0.016}$ |
| LCMD+STAP | $-0.138_{\pm 0.017}$ | $-1.561_{\pm 0.016}$ | $-2.033_{\pm 0.017}$ | $1.139_{\pm 0.006}$ | $0.841_{\pm 0.014}$ | $-0.431_{\pm 0.016}$ |

We provide a full report of all results from the main experiment. Table 8, Table 9, Table 10, Table 11, and Table 12 show the full results on Burgers, KdV, KS, and NS equations, respectively. Fig. 12 shows the plots of RMSE quantiles on all PDEs.

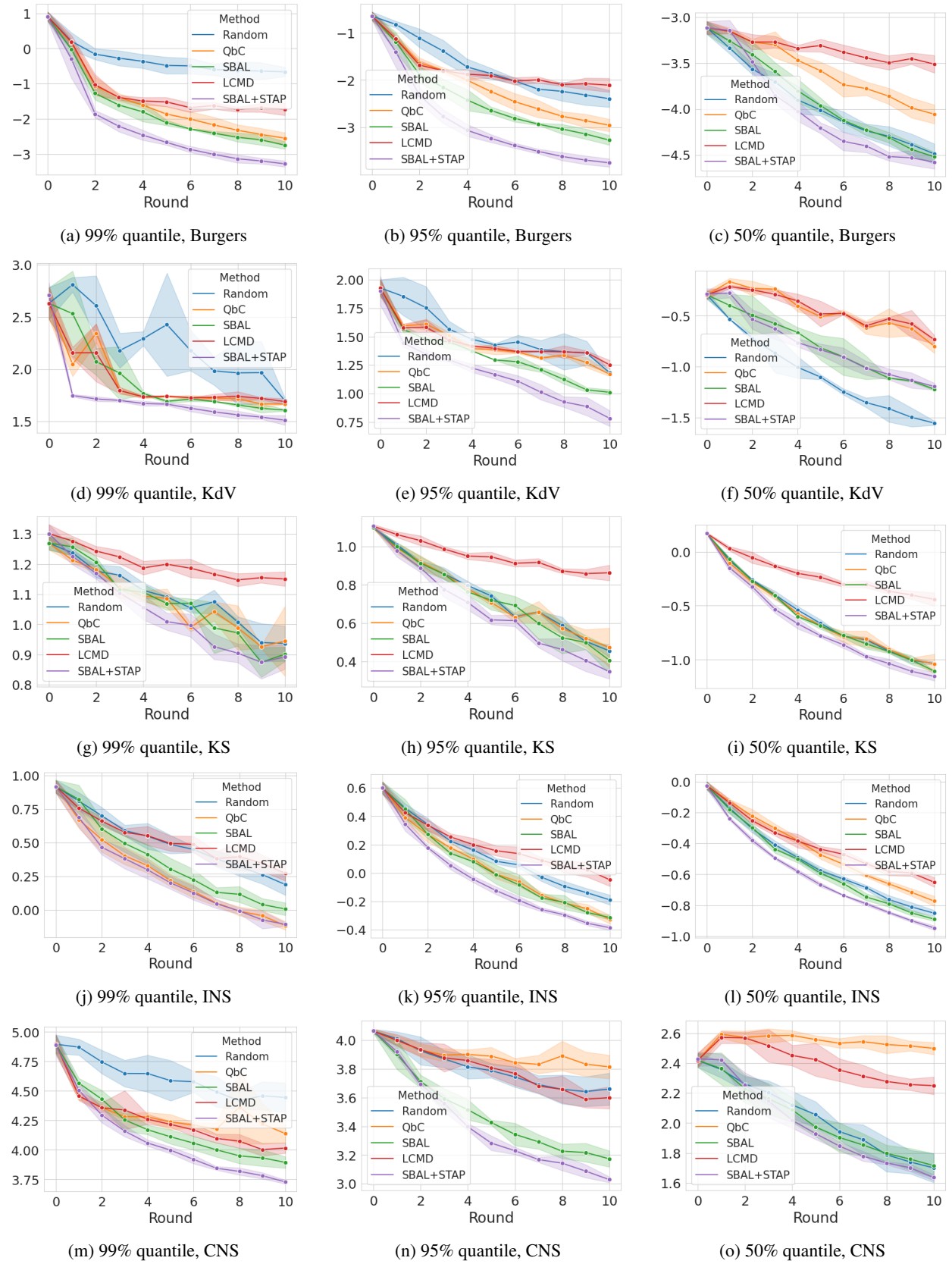

Figure 12: Log RMSE quantiles

Table 11: Mean log metrics for INS Equation

| | RMSE | NRMSE | MAE | 99% | 95% | 50% |
|---|---|---|---|---|---|---|
| Random | $-0.422_{\pm 0.010}$ | $-2.714_{\pm 0.010}$ | $-2.698_{\pm 0.010}$ | $0.515_{\pm 0.021}$ | $0.134_{\pm 0.012}$ | $-0.520_{\pm 0.010}$ |
| Random+STAP | $-0.489_{\pm 0.012}$ | $-2.780_{\pm 0.012}$ | $-2.764_{\pm 0.012}$ | $0.412_{\pm 0.042}$ | $0.037_{\pm 0.025}$ | $-0.581_{\pm 0.010}$ |
| SBAL | $-0.461_{\pm 0.012}$ | $-2.751_{\pm 0.012}$ | $-2.736_{\pm 0.009}$ | $0.372_{\pm 0.039}$ | $0.043_{\pm 0.024}$ | $-0.543_{\pm 0.014}$ |
| SBAL+STAP | $-0.525_{\pm 0.005}$ | $-2.814_{\pm 0.005}$ | $-2.797_{\pm 0.003}$ | $0.268_{\pm 0.027}$ | $-0.043_{\pm 0.007}$ | $-0.601_{\pm 0.003}$ |
| QbC | $-0.385_{\pm 0.014}$ | $-2.675_{\pm 0.015}$ | $-2.663_{\pm 0.016}$ | $0.282_{\pm 0.024}$ | $0.047_{\pm 0.013}$ | $-0.439_{\pm 0.015}$ |
| QbC+STAP | $-0.466_{\pm 0.011}$ | $-2.757_{\pm 0.011}$ | $-2.746_{\pm 0.010}$ | $0.205_{\pm 0.021}$ | $-0.036_{\pm 0.014}$ | $-0.518_{\pm 0.010}$ |
| LCMD | $-0.320_{\pm 0.011}$ | $-2.607_{\pm 0.010}$ | $-2.587_{\pm 0.010}$ | $0.532_{\pm 0.019}$ | $0.202_{\pm 0.011}$ | $-0.400_{\pm 0.012}$ |
| LCMD+STAP | $-0.378_{\pm 0.012}$ | $-2.665_{\pm 0.012}$ | $-2.645_{\pm 0.011}$ | $0.476_{\pm 0.017}$ | $0.145_{\pm 0.020}$ | $-0.462_{\pm 0.011}$ |

Table 12: Mean log metrics for CNS Equation

| | RMSE | NRMSE | MAE | 99% | 95% | 50% |
|---|---|---|---|---|---|---|
| Random | $2.603_{\pm 0.038}$ | $-2.498_{\pm 0.050}$ | $-0.732_{\pm 0.031}$ | $4.618_{\pm 0.057}$ | $3.806_{\pm 0.050}$ | $2.044_{\pm 0.050}$ |
| Random+STAP | $2.461_{\pm 0.017}$ | $-2.637_{\pm 0.015}$ | $-0.863_{\pm 0.013}$ | $4.458_{\pm 0.057}$ | $3.659_{\pm 0.020}$ | $1.905_{\pm 0.030}$ |
| SBAL | $2.422_{\pm 0.045}$ | $-2.482_{\pm 0.040}$ | $-0.880_{\pm 0.041}$ | $4.206_{\pm 0.038}$ | $3.497_{\pm 0.044}$ | $2.024_{\pm 0.064}$ |
| SBAL+STAP | $2.363_{\pm 0.016}$ | $-2.501_{\pm 0.051}$ | $-0.932_{\pm 0.015}$ | $4.096_{\pm 0.019}$ | $3.423_{\pm 0.018}$ | $1.990_{\pm 0.031}$ |
| QbC | $2.844_{\pm 0.021}$ | $-2.102_{\pm 0.024}$ | $-0.476_{\pm 0.018}$ | $4.329_{\pm 0.045}$ | $3.899_{\pm 0.022}$ | $2.539_{\pm 0.015}$ |
| QbC+STAP | $2.788_{\pm 0.017}$ | $-2.063_{\pm 0.021}$ | $-0.534_{\pm 0.015}$ | $4.249_{\pm 0.017}$ | $3.783_{\pm 0.010}$ | $2.518_{\pm 0.022}$ |
| LCMD | $2.736_{\pm 0.029}$ | $-2.227_{\pm 0.028}$ | $-0.582_{\pm 0.029}$ | $4.263_{\pm 0.021}$ | $3.803_{\pm 0.032}$ | $2.401_{\pm 0.032}$ |
| LCMD+STAP | $2.739_{\pm 0.002}$ | $-2.147_{\pm 0.009}$ | $-0.572_{\pm 0.002}$ | $4.143_{\pm 0.017}$ | $3.726_{\pm 0.017}$ | $2.478_{\pm 0.015}$ |

# D. Additional experiments

## D.1. Active learning with 20 Rounds

We have performed the main experiment for 20 rounds instead of 10, on Random, SBAL, and SBAL+STAP. The results are shown in Fig. 14. We observe that the gap between SBAL and SBAL+STAP keeps widening, except in the KdV equation.

## D.2. Diversity of Sparsely Selected Time Steps

We provide a simple analysis to show that time steps sampled in a sparse manner are more diverse than time steps from entire trajectories. Out of 128 trajectories, we first randomly chose 10 trajectories, which contains $L \times 10$ states. Then, out of all $L \times 128$ states, we randomly chose $L \times 10$ states. The first choice represents full trajectory sampling, and the latter represents spare time steps sampling. We probe an FNO surrogate model trained on all the 128 trajectories at its hidden layer, and observe the hidden layer activation at each of the $L \times 128$ states. The result is shown in Fig. 15, where black points represent states from the fully sampled trajectories and red points represent sparsely selected states. The latter states are visibly more diverse, which partially explains how sampling time steps in a sparse manner from trajectories can benefit a surrogate model.

## D.3. Random Bernoulli Sampling of Time Steps

We provide the whole list of results with Bernoulli sampling described in § 5.5. Also, we can enforce consecutive initial time steps sampling by bringing all the true entries in $S$ to the beginning. We call this method Initial Bernoulli sampling, or Initial Ber($p$). We report the results with SBAL in Table 13 and Table 14. Initial Bernoulli sampling always performs the worst, possibly because they rarely see the time steps at the end.

## D.4. Time-dependent Incompressible Navier-Stokes

We have performed an experiment on a time-dependent incompressible Navier Stokes equation, simply by using the time-dependent external force in our current INS equation. The new forcing term is a sinusoidal mixture of two spatial coordinates and the temporal coordinate. Fig. 16 shows the log RMSE of Random, SBAL, and SBAL+STAP. Since our methodology aligns closely to the time-dependent PDEs, our method significantly outperforms the others.

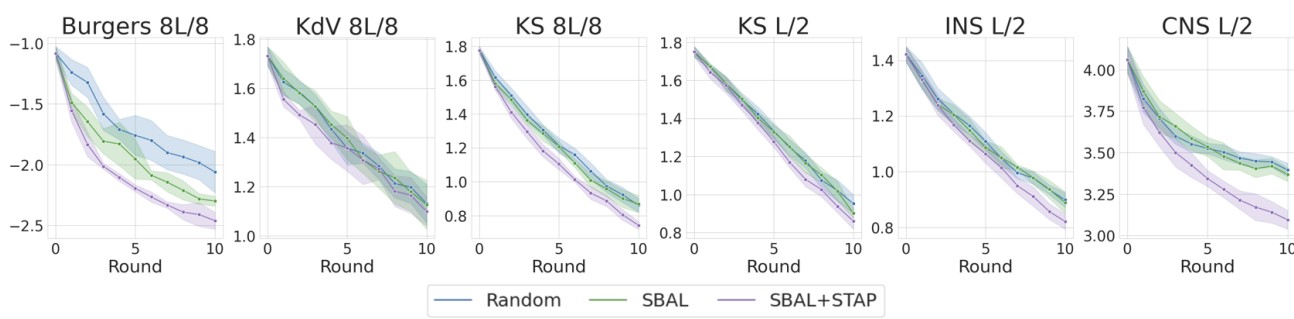

Figure 13: Log RMSE of AL strategies with multi-step FNO, across 10 rounds of acquisition.

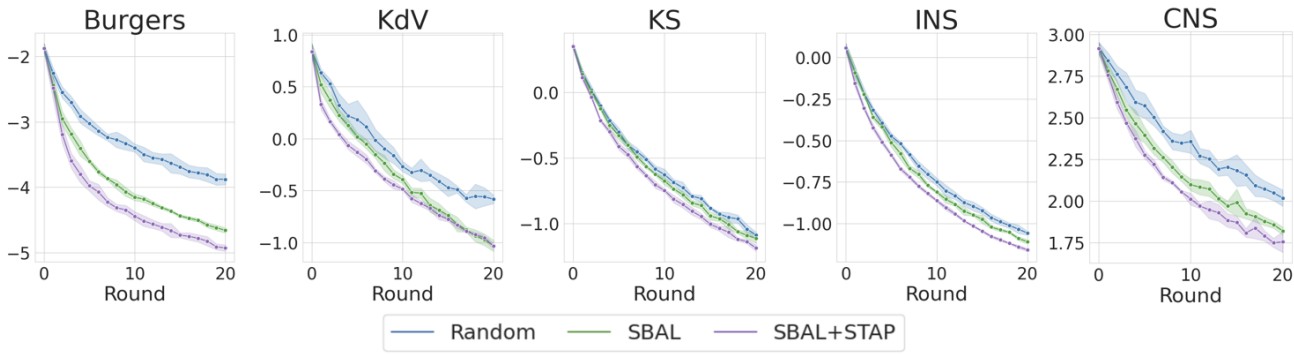

Figure 14: Log RMSE of AL strategies, measured across 20 rounds of acquisition.

## E. Comparison of Costs with Our Numerical Solvers

Our paper does not claim direct computational speedups on our benchmark PDEs; instead, it relies on the benchmark PDEs as proxies reflecting realistic, expensive simulations. Our surrogate metric, the number of numerical solver-simulated timesteps, effectively represents relative computational savings in realistic, expensive simulations.

However, we provide the analysis of computational cost on our benchmark PDEs for the sake of completeness. We compare the wall clock time of Random and SBAL+STAP, and for a meaningful comparison, we cut the SBAL+STAP experiment when it reaches below the RMSE of the final Random surrogate model.

Table 15 compares the wall clock times between non-AL and AL, and Table 16 decomposes them into data acquisition, model training, and data selection. We find that AL reduces the total cost in CNS, where acquisition is relatively expensive. On other benchmarks, the training and data selection costs dominate, as expected. We want to stress yet again that the benchmarks were intentionally chosen to be inexpensive, to enable fast experimentation.

We provide the condition under which AL reduces total cost. Suppose AL improves data efficiency by $E$ over non-AL. Define $T_{\text{acquire}}, T_{\text{train}}$ as the acquisition time and training time per unit data, and $T_{\text{select}}$ as the data selection time per round. The total cost of non-AL is

$$N_{\text{acquire}}^{(1)} T_{\text{acquire}} + N_{\text{train}}^{(1)} T_{\text{train}}$$

and for AL,

$$N_{\text{acquire}}^{(2)} T_{\text{acquire}} + N_{\text{train}}^{(2)} T_{\text{train}} + M T_{\text{select}}$$

where $N_{\text{acquire}}^{(i)}$ are the number of acquired data, and $N_{\text{train}}^{(i)}$ are the total number of training examples (counting duplicates),

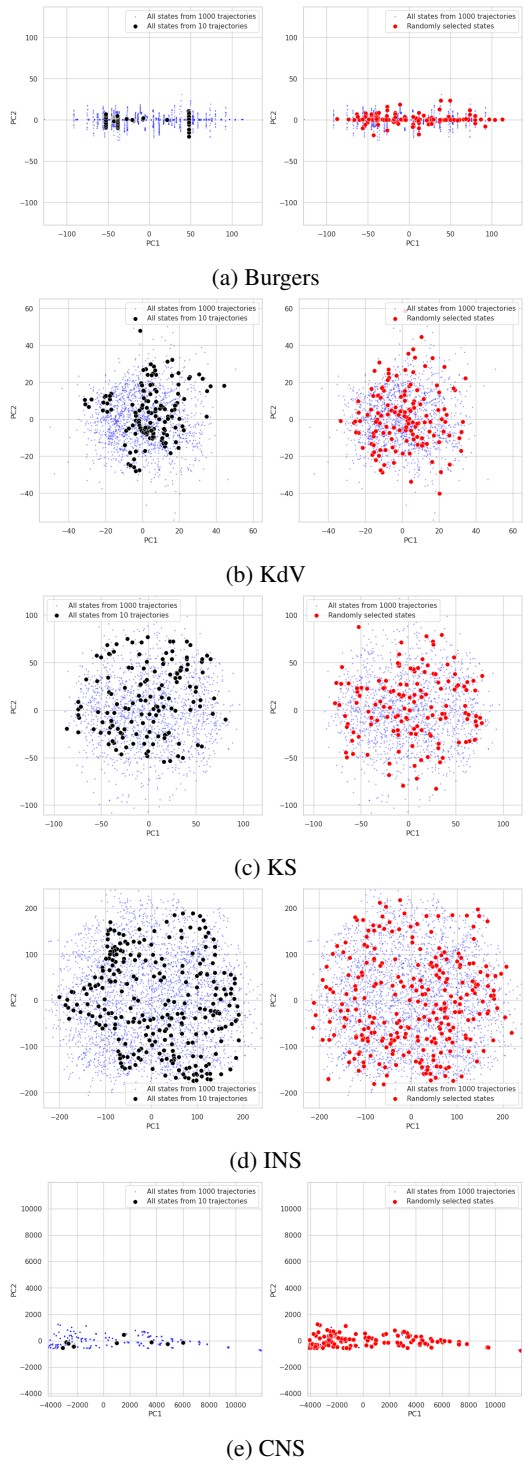

Figure 15: PCA of FNO hidden layer's activation pattern for both entire trajectories (black) and sparsely sampled time steps (red)

Table 13: Bernoulli sampling

|  | SBAL | +STAP | +Ber(1/16) | +Ber(1/8) | +Ber(1/4) | +Ber(1/2) |
|---|---|---|---|---|---|---|
| **Burgers** | | | | | | |
| RMSE | $-3.388_{\pm0052}$ | $-3.674_{\pm0.071}$ | $-3.231_{\pm0.163}$ | $-3.152_{\pm0.267}$ | $-3.102_{\pm0.441}$ | $-3.458_{\pm0.067}$ |
| NRMSE | $-4.081_{\pm0.061}$ | $-4.306_{\pm0.065}$ | $-3.847_{\pm0.171}$ | $-3.782_{\pm0.259}$ | $-3.734_{\pm0.447}$ | $-4.122_{\pm0.078}$ |
| MAE | $-5.332_{\pm0.046}$ | $-5.585_{\pm0.066}$ | $-5.169_{\pm0.164}$ | $-5.085_{\pm0.272}$ | $-5.037_{\pm0.442}$ | $-5.397_{\pm0.063}$ |
| **KdV** | | | | | | |
| RMSE | $0.030_{\pm0.029}$ | $-0.088_{\pm0.040}$ | $0.053_{\pm0.014}$ | $0.049_{\pm0.014}$ | $0.018_{\pm0.024}$ | $-0.064_{\pm0.031}$ |
| NRMSE | $-1.282_{\pm0.030}$ | $-1.378_{\pm0.040}$ | $-1.254_{\pm0.017}$ | $-1.257_{\pm0.014}$ | $-1.288_{\pm0.020}$ | $-1.370_{\pm0.033}$ |
| MAE | $-2.139_{\pm0.027}$ | $-2.239_{\pm0.043}$ | $-2.082_{\pm0.016}$ | $-2.083_{\pm0.018}$ | $-2.120_{\pm0.025}$ | $-2.207_{\pm0.034}$ |
| **KS** | | | | | | |
| RMSE | $-0.275_{\pm0.014}$ | $-0.349_{\pm0.003}$ | $-0.365_{\pm0.008}$ | $-0.359_{\pm0.006}$ | $-0.346_{\pm0.008}$ | $-0.324_{\pm0.007}$ |
| NRMSE | $-1.700_{\pm0.014}$ | $-1.774_{\pm0.003}$ | $-1.790_{\pm0.008}$ | $-1.784_{\pm0.006}$ | $-1.771_{\pm0.008}$ | $-1.749_{\pm0.007}$ |
| MAE | $-2.184_{\pm0.014}$ | $-2.265_{\pm0.003}$ | $-2.282_{\pm0.007}$ | $-2.276_{\pm0.006}$ | $-2.262_{\pm0.009}$ | $-2.237_{\pm0.009}$ |
| **INS** | | | | | | |
| RMSE | $-0.422_{\pm0.010}$ | $-0.525_{\pm0.005}$ | $-0.529_{\pm0.006}$ | $-0.525_{\pm0.006}$ | $-0.515_{\pm0.007}$ | $-0.500_{\pm0.009}$ |
| NRMSE | $-2.751_{\pm0.012}$ | $-2.814_{\pm0.005}$ | $-2.818_{\pm0.006}$ | $-2.814_{\pm0.006}$ | $-2.805_{\pm0.007}$ | $-2.789_{\pm0.009}$ |
| MAE | $-2.736_{\pm0.009}$ | $-2.797_{\pm0.003}$ | $-2.794_{\pm0.006}$ | $-2.790_{\pm0.006}$ | $-2.781_{\pm0.007}$ | $-2.769_{\pm0.008}$ |
| **CNS** | | | | | | |
| RMSE | $2.422_{\pm0.045}$ | $2.363_{\pm0.016}$ | $2.375_{\pm0.069}$ | $2.370_{\pm0.018}$ | $2.372_{\pm0.012}$ | $2.392_{\pm0.012}$ |
| NRMSE | $-2.482_{\pm0.040}$ | $-2.501_{\pm0.051}$ | $-2.540_{\pm0.058}$ | $-2.526_{\pm0.024}$ | $-2.521_{\pm0.021}$ | $-2.519_{\pm0.013}$ |
| MAE | $-0.880_{\pm0.041}$ | $-0.932_{\pm0.015}$ | $-0.921_{\pm0.064}$ | $-0.925_{\pm0.019}$ | $-0.923_{\pm0.012}$ | $-0.906_{\pm0.010}$ |

and $M$ the number of rounds. With initial datasize $D$ and acquired datasize $B$ per round,

$$N^{(1)}_{\text{acquire}} = BM$$

$$N^{(1)}_{\text{train}} = D + BM$$

$$N^{(2)}_{\text{acquire}} = BM/E$$

$$N^{(2)}_{\text{train}} = \sum_{\text{round}=0}^{M/E} (D + B \cdot \text{round})$$

For AL to reduce the total cost, the setting would need to satisfy

$$N^{(1)}_{\text{acquire}} T_{\text{acquire}} + N^{(1)}_{\text{train}} T_{\text{train}} > N^{(2)}_{\text{acquire}} T_{\text{acquire}} + N^{(2)}_{\text{train}} T_{\text{train}} + M T_{\text{select}}$$

Table 17 lists these values, and whether they satisfy the condition above.

Table 14: Initial Bernoulli sampling

|  | Initial Ber(1/16) | Initial Ber(1/8) | Initial Ber(1/4) | Initial Ber(1/2) |
|---|---|---|---|---|
| **Burgers** | | | | |
| RMSE | $-3.737_{\pm 0.034}$ | $-3.686_{\pm 0.064}$ | $-3.634_{\pm 0.024}$ | $-3.551_{\pm 0.043}$ |
| NRMSE | $-4.368_{\pm 0.049}$ | $-4.322_{\pm 0.065}$ | $-4.274_{\pm 0.027}$ | $-4.222_{\pm 0.056}$ |
| MAE | $-5.654_{\pm 0.027}$ | $-5.607_{\pm 0.058}$ | $-5.561_{\pm 0.018}$ | $-5.483_{\pm 0.034}$ |
| **KdV** | | | | |
| RMSE | $0.032_{\pm 0.016}$ | $-0.015_{\pm 0.014}$ | $-0.001_{\pm 0.018}$ | $0.011_{\pm 0.014}$ |
| NRMSE | $-1.278_{\pm 0.014}$ | $-1.321_{\pm 0.017}$ | $-1.303_{\pm 0.018}$ | $-1.294_{\pm 0.014}$ |
| MAE | $-2.150_{\pm 0.016}$ | $-2.197_{\pm 0.014}$ | $-2.181_{\pm 0.018}$ | $-2.168_{\pm 0.014}$ |
| **KS** | | | | |
| RMSE | $-0.302_{\pm 0.009}$ | $-0.293_{\pm 0.008}$ | $-0.287_{\pm 0.005}$ | $-0.283_{\pm 0.009}$ |
| NRMSE | $-1.728_{\pm 0.009}$ | $-1.719_{\pm 0.008}$ | $-1.713_{\pm 0.005}$ | $-1.708_{\pm 0.009}$ |
| MAE | $-2.216_{\pm 0.008}$ | $-2.206_{\pm 0.008}$ | $-2.199_{\pm 0.007}$ | $-2.194_{\pm 0.010}$ |
| **INS** | | | | |
| RMSE | $-0.282_{\pm 0.005}$ | $-0.294_{\pm 0.010}$ | $-0.329_{\pm 0.005}$ | $-0.397_{\pm 0.008}$ |
| NRMSE | $-2.575_{\pm 0.005}$ | $-2.587_{\pm 0.010}$ | $-2.621_{\pm 0.005}$ | $-2.688_{\pm 0.008}$ |
| MAE | $-2.597_{\pm 0.004}$ | $-2.609_{\pm 0.009}$ | $-2.643_{\pm 0.004}$ | $-2.703_{\pm 0.007}$ |
| **CNS** | | | | |
| RMSE | $2.431_{\pm 0.017}$ | $2.420_{\pm 0.009}$ | $2.389_{\pm 0.011}$ | $2.420_{\pm 0.009}$ |
| NRMSE | $-2.492_{\pm 0.020}$ | $-2.493_{\pm 0.011}$ | $-2.545_{\pm 0.009}$ | $-2.501_{\pm 0.013}$ |
| MAE | $-0.873_{\pm 0.015}$ | $-0.883_{\pm 0.008}$ | $-0.918_{\pm 0.009}$ | $-0.886_{\pm 0.008}$ |

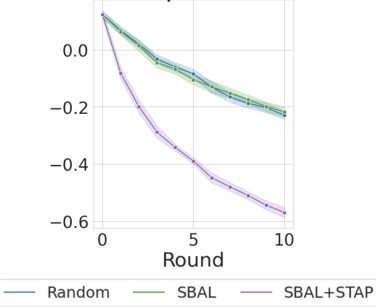

Figure 16: Log RMSE of AL strategies on time-dependent INS.

Table 15: Total wall clock time until target RMSE (in seconds)

| Equation | Random | SBAL+STAP |
|---|---|---|
| Burgers | 237 | 441 |
| KdV | 834 | 1469 |
| KS | 385 | 2920 |
| INS | 526 | 3702 |
| CNS | 4053 | 3476 |

Table 16: Total wall clock time decomposed into acquisition/training/selection (in seconds)

| Equation | Random | SBAL+STAP |
|---|---|---|
| Burgers | 90/147/0 | 27/234/180 |
| KdV | 670/164/0 | 455/664/350 |
| KS | 40/345/0 | 35/2075/810 |
| INS | 190/336/0 | 160/1750/1792 |
| CNS | 3570/483/0 | 1448/972/1056 |

Table 17: Variables for cost analysis

| Equation | $E$ | $T_{\text{acquire}}$ | $T_{\text{train}}$ | $T_{\text{select}}$ | Satisfied |
|---|---|---|---|---|---|
| Burgers | 3.33 | 0.087 | 0.101 | 60 | F |
| KdV | 1.43 | 0.654 | 0.106 | 50 | F |
| KS | 1.11 | 0.005 | 0.116 | 90 | F |
| INS | 1.25 | 0.077 | 0.112 | 224 | F |
| CNS | 2.50 | 1.760 | 0.157 | 264 | T |

