# OpenReview forum: "Active Learning with Selective Time-Step Acquisition for PDEs"
_ICML.cc/2025/Conference — ICML 2025 poster_

### Official Review · Reviewer_K9ac · 2025-03-11

**Overall Recommendation:** 3

**Summary:**

This paper introduced an active learning method for learning PDEs. The method is composed of: (1) selective time-step acquisition, where the method selects a subset of time steps for the solver to simulate while other time steps are evolved by the surrogate model, (2) an acquisition function that evaluates the variance reduction, and (3) batch acquisition. The method is evaluated on 5 PDEs including incompressible and compressible NS, KS, Burgers, and KdV equations. The method shows consistent improvement compared with other baselines.

**Claims And Evidence:**

Yes. The claims that the selective time-step acquisition improves upon full trajectory acquisition is supported by convincing evidence.

**Essential References Not Discussed:**

Not that I know of.

**Experimental Designs Or Analyses:**

The experimental design is sound. Experiments are done on incompressible and compressible NS equations which are important evaluations. Also, experiments are done comparing the method with random Bernoulli sampling, which shows that although the proposed method is on par with best Bernoulli sampling result.

I wonder if the patterns shown in Figure 5 are consistent among different experiment runs (with different seeds)? The authors are encouraged to run multiple independent experiments, showing their patterns.

**Methods And Evaluation Criteria:**

Yes, they make sense

**Other Comments Or Suggestions:**

N/A

**Other Strengths And Weaknesses:**

Strengths:

The paper is written in a clear way and easy to understand. The novelty is reasonable. In terms of significance, I believe the paper addresses an important problem that can be widely used in learning PDEs.

Weaknesses:

The thing I worry about most is the engineering difficulty of applying the method. The method requires interleaving solver evolution with surrogate evolution, which, in my own experience, is typically engineering challenging. For example, if the solver is a well-established software which is written in a different language, it can be very hard to incorporate it with a neural network. In those case, it is probably easier if the solver runs the full trajectory. Therefore, in practice, I wonder whether the gain by selective time-step acquisition is worth the effort. Nevertheless, when incorporating the two component is not too hard, the method may be useful.

## update after rebuttal:
I have read the authors' rebuttal and mostly resolved my concerns. I remain my score.

**Questions For Authors:**

N/A

**Relation To Broader Scientific Literature:**

The paper addresses an important problem in learning PDEs, where the time cost for running the solver is high.

**Theoretical Claims:**

The paper does not have theoretical claims.

---

> ### Author Rebuttal · Authors · 2025-04-01
>
> We appreciate your thoughtful questions and feedback.
>
> > I wonder if the patterns shown in Figure 5 are consistent among different experiment runs (with different seeds)? The authors are encouraged to run multiple independent experiments, showing their patterns.
>
> https://anonymous.4open.science/r/icml_rebuttal-E9AB/timesteps/
>
> Thank you for raising an important point. The linked folder "timesteps" contains sampled timesteps for four other seeds for each of the PDEs. We see that the patterns described in our paper are consistent among different seeds per task.
>
> > The method requires interleaving solver evolution with surrogate evolution, which, in my own experience, is typically engineering challenging.
>
> We appreciate the concern about interleaving solver-based and surrogate-based time steps. In our current codebase, the only required addition is calling one step of the external solver from Python. In practice, most PDE solvers can evolve a state one step at a time through a simple function call or by using “checkpoint” files between time steps, which Python can read and write. Although some solvers are specialized or written in lower-level languages, the overhead incurred by periodic file-based communication is generally much smaller than the total cost of running a full solver trajectory, especially for problems where the solver is very expensive. By selectively acquiring just the most important steps, our approach saves substantial solver calls and overall runtime. We believe the net cost reduction outweighs the implementation complexity in most scenarios where computational demands are high. We will also make sure that our codebase is generally easy to adopt for user-defined PDE solvers.

---

### Official Review · Reviewer_f7JP · 2025-03-11

**Overall Recommendation:** 4

**Summary:**

This paper introduces a novel active learning (AL) framework called Selective Time-Step Acquisition for PDEs (STAP) to improve the efficiency of surrogate models for partial differential equations (PDEs). The key idea is to selectively acquire only the most informative time steps from PDE trajectories using a numerical solver while the surrogate model approximates the remaining steps. This contrasts with existing AL methods that acquire entire PDE trajectories, reducing computational costs per trajectory and allowing for exploring a more diverse set of trajectories within a given budget. The authors develop an acquisition function that estimates the utility of a set of time steps by approximating the resulting variance reduction. They demonstrate the effectiveness of STAP on benchmark PDEs like the Burgers', Korteweg-De Vries, Kuramoto-Sivashinsky, incompressible Navier-Stokes, and compressible Navier-Stokes equations. Results show that STAP significantly improves performance, reducing average error and error quantiles compared to existing methods. The method combines a numerical solver and a surrogate model to acquire data along a trajectory with reduced cost, improving sample efficiency. STAP can be seen as an add-on to existing AL methods that acquire full trajectories. The authors also explore efficient variants of STAP to reduce computational cost, and show that the success of STAP is driven by its ability to prioritize both diverse and informative time steps. The findings suggest STAP offers a more cost-efficient and accurate solution for PDE surrogate modeling with broader applicability in scientific and engineering simulations.

**Claims And Evidence:**

Claim: STAP improves surrogate model performance over previous active learning methods for PDEs.

Evidence: Experiments on Burgers', KdV, KS, INS, and CNS equations demonstrate that SBAL+STAP outperforms other AL baselines regarding Log RMSE (Figure 3, Table 1). "Ratio of ∆" values in Table 1 quantifies the improvement of SBAL+STAP over Random selection. These ratios are greater than 1 for all tested PDEs, indicating STAP provides a performance gain. Figure 4 shows that STAP improves performance on points with extreme errors (99% and 95% quantiles) and even reduces error in the middle quantiles (50%), which is rare for AL algorithms. Appendix C provides a full report of all metrics on all methods

Claim: STAP achieves better performance by adaptively choosing both the frequency and specific locations of time steps to acquire.

Evidence: Figure 5 shows the distribution of time steps chosen by STAP, demonstrating a tendency to acquire early time steps, with occasional selection of later time steps varying across different PDEs.  Analysis of the frequency with which STAP samples time steps reveals that it doesn't always sample time steps at an optimal frequency p to sample with. This suggests that the specific time steps sampled also matter as much as the overall frequency.

Claim: STAP can be implemented efficiently without significant performance loss.

Evidence: The use of two efficient variants of STAP, namely STAP MF and STAP 10, shown in Table 14.
Table 3 summarizes the wall-clock time of each baseline method and STAP; STAP 10 incurs only a fraction of computational cost over the baseline methods.

**Essential References Not Discussed:**

None.

**Experimental Designs Or Analyses:**

The experiments are well-designed and in line with prior work by Musekamp et al. but limited to one-step predictions.

**Methods And Evaluation Criteria:**

The paper uses a well-chosen set of PDE benchmarks from the Al4PDE active learning benchmark, including Burgers', KdV, KS, INS, and CNS equations, which represent a variety of physical phenomena and complexity levels, making for a robust and generalizable evaluation of STAP. It is great that the authors use the existing AL benchmarks and it would be great if they could also publish their method on github and integrate it there with the existing framework.

**Other Comments Or Suggestions:**

None.

**Other Strengths And Weaknesses:**

Strengths
I really like the PCA visualizations of the effect. It is impressive that the methods show improvements over random sampling, even for problems where no other AL method showed an improvement!  The evaluation is strong and on multiple PDEs from AL4PDE.


Weakness
The method could be prone to diverging models, producing out-of-distribution inputs to the simulator. There is no check of the inputs in terms of the uncertainty.

Only applicable to 1-step training (predicting t+1 from t), does not support autoregressive training techniques.

A push-forward experiment (D.2) is a good idea to check. But the numbers seem wrong; it is hard to believe that the push-forward trick worsens things.

The connection to PDE dynamics / autoregressive behavior could have been discussed more. For example, your algorithm likely selects earlier time steps since improvements, in the beginning, will affect the rest of the rollout.  In the autoregressive setting, one might not want to select only the first steps since it will take some time for the model to diverge again and since they might be too close.

**Questions For Authors:**

What does the +- / shaded area show exactly? Confidence interval? 1 time standard deviation?  The standard deviation was probably underestimated since they took the average error of the ensemble members (ln 848).

Retraining == training from scratch or fine-tuning?

What are the parameters of the IC generator (distribution)?

**Relation To Broader Scientific Literature:**

Active learning for ML-based PDE solvers is a recent development, with only a few recent papers having addressed this important problem. The authors make strong contributions by showing that time steps of the trajectories (rollouts) can be chosen adaptively and selectively. Compared to prior work, this is a significant advancement over the state of the art.

**Theoretical Claims:**

There are no theoretical claims made in the paper.

---

> ### Author Rebuttal · Authors · 2025-04-01
>
> We appreciate your thoughtful questions and feedback.
>
> > Prone to diverging models, producing out-of-distribution inputs
>
> Thank you for raising this important point. Please see Common Response 2 at the bottom.
>
> > Does not support autoregressive training techniques
>
> We have done additional experiments with multi-step models as in [1]. Please see Common Response 1 in our response to reviewer FkHn.
>
> > What does the +- / shaded area show exactly?
>
> We used one standard deviation. You are correct to point out that we underestimated the standard deviation. We will rerun the experiments to obtain the correct standard deviations with prediction error of all ensemble members across all seeds.
>
> https://anonymous.4open.science/api/repo/icml_rebuttal-E9AB/file/correct_std.png?v=273fbf72
>
> The above figure contains the corrected log RMSE figure for some main experiments.
>
> > Retraining == ?
>
> We are training from scratch like in most of the active learning literature [1].
>
> > Parameters of IC generators
>
> Let $U$ stand for the uniform distribution. For Burgers and KdV, the initial condition is in the form $\sum_{i=1}^N A_i \sin(2\pi k_i x / L + \phi_i)$. The amplitudes and phases are always sampled from $U([0,1])$ and $U([0,2\pi])$. For Burgers, $N = 2$ and $k_i \sim U(\\{1,2,3,4\\})$, and for KdV, $N=10$ and $k_i \sim U(\\{1,2,3\\})$. For KS and INS, the initial conditions are gaussian random fields drawn from $N(0, 25(-\Delta + 25I)^{-1})$ and $N(0, 7^{3/2}(-\Delta + 49I)^{-2.5})$ respectively. For CNS, we first sample $\sum_{k \in \\{1,2,3,4\\}^3} A_k \sin( 2\pi k x/L + \phi_k)$, the amplitudes and phases sampled uniformly at random for each channel ($\rho$, $p$ and $\mathbf{v}$). Then, we renormalize $\rho$, $p$ to lie within $\rho_0(1 \pm \Delta_\rho)$ and $p_0(1 \pm \Delta_p)$, respectively, where $\rho_0 \sim U([0.1,10])$, $\Delta_\rho \sim U([0.013, 0.26])$, $\Delta_p \sim U([0.04,0.8])$ and $T_0 := \rho_0/p_0 \sim U([0.1,10])$. The $\mathbf{v}$ is also computed by superposing sinusoidal waves, but with amplitudes chosen so that the initial condition has the given initial Mach number $M \sim U([0.1,1])$.
>
> > Push-forward trick
>
> We agree that the errors seem too high. If we can’t fix the problem by the deadline, we will remove the push-forward experiment.
>
> **Common Response 2. Out-of-distribution Synthetic Inputs**
>
> https://anonymous.4open.science/r/icml_rebuttal-E9AB/gt_pred/ \
> https://anonymous.4open.science/api/repo/icml_rebuttal-E9AB/file/kdv_energy.png?v=5d240a44
>
> Reviewers have raised the concern that inaccurate surrogate models might synthesize inputs that lie far from the ground truth distribution, reducing their information gain. In fact, under limited training data, the surrogate model outputs visibly erroneous trajectories. The images in the folder of the first link are comparisons of the ground truth trajectories and predictions from surrogate models trained on 32 and 1 trajectories, respectively. The second link plots the energy of KdV states in both ground truth and predicted trajectories. We see that the surrogate model doesn't satisfy conservation of energy.
>
> https://anonymous.4open.science/api/repo/icml_rebuttal-E9AB/file/one_initial_train.png?v=1d82126d \
> https://anonymous.4open.science/r/icml_rebuttal-E9AB/stap_pca/
>
> To test how much this error harms STAP, we perform experiments where the initial training dataset contains 1 trajectory, compared to 32 used in our main experiments. The first link above shows the log RMSEs. To our own surprise, we find that SBAL+STAP still outperforms Random and SBAL, except for INS in the early rounds.
> The second link is a folder that contains comparisons of FNO activation PCAs for PDE states sampled by SBAL and SBAL+STAP during the first round of active learning. Note that the PCA was fitted only to the ground truth states in random trajectories (blue points), so that the out-of-distributionness of sampled states can be properly reflected. We find that only several of the states sampled by SBAL+STAP diverge significantly from the random ground truth states. In other words, the surrogate model synthesizes erroneous inputs when looked at trajectory-wise, but they aren't necessarily out-of-distribution, hence retaining the information gain.
>
> Reviewer vQAV also asked, "why not query more diverse initial conditions and run fewer time steps" in the earlier rounds where the surrogate model is inaccurate. The problem is that the distribution of states $u_t$ changes over time $t$. Running only up to the first few timesteps harms the model’s performance on the later timesteps, as evidenced in Appendix D.3. STAP seems to be striking the balance between sampling realistic inputs and sampling diverse timesteps $t$. How one could further improve this balance is left as a work for future research.
>
> References:
>
> [1] Musekamp, Daniel, et al. "Active learning for neural pde solvers." arXiv preprint arXiv:2408.01536 (2024).

---

### Official Review · Reviewer_FkHn · 2025-03-13

**Overall Recommendation:** 2

**Summary:**

The paper introduces an active learning framework STAP for surrogate modeling of PDE trajectories that selectively queries only key time steps instead of simulating entire trajectories. STAP uses a binary sampling pattern to decide which time steps to acquire via a numerical solver and which to approximate with a surrogate model. An acquisition function based on variance reduction guides the selection process, and a greedy batch acquisition algorithm is used to optimize the sampling patterns under a fixed-cost budget. Experiments on benchmark PDEs, including Burgers, KdV, Kuramoto-Sivashinsky, and Navier-Stokes equations, demonstrate that the proposed method reduces errors compared to existing active learning methods, particularly at higher quantiles.

## update after rebuttal

I thank the authors for the updated results. I appreciate the interesting idea presented in the paper. However, although aiming to reduce the cost of generating a PDE dataset, the method presented in this paper, in the end, makes the overall process of obtaining a surrogate model more expensive in most cases. I believe the paper can contribute validly to the field if the authors address this contradiction. I've decided to maintain my score since two specific issues regarding the evidence presented in this manuscript remain unaddressed.

Total computational cost is larger: In the AI4PDE context, both data generation (numerical solvers) and model training/AL overhead consume computational resources. Therefore, demonstrating a reduction in total computational cost is crucial for establishing practical value. While the authors provided cost analysis upon request, the results indicated their method actually increased the total computational cost for most benchmarks presented. If the proposed AL method requires more total computation time or resources to reach a given accuracy on these problems, what is its compelling advantage over standard approaches? The paper needs to address this directly and convincingly, rather than solely relying on hypothetical scenarios.

Unvalidated scalability assumption for larger-scale PDEs: The work's significance heavily relies on extrapolating its findings to more complex, high-dimensional PDEs where solver costs are presumed dominant. However, it does not consider whether the deep learning surrogate, along with the active learning strategy that depends on it, can effectively scale to these types of problems. This is a non-trivial assumption, especially in light of known limitations of deep learning, such as producing blurry predictions for very high-resolution data. Without supporting evidence, expectations regarding performance in complex regimes remain speculative.

**Claims And Evidence:**

The paper claims that selective time-step acquisition using active learning improves the efficiency and accuracy of PDE surrogate modeling by reducing the expensive computation of the numerical solver. While the experimental results provide clear evidence that the proposed method outperforms other active learning methods, no cost of time and computing resources analysis against direct non-active learning are provided. The active learning in this work consists of training an ensemble of FNOs over multiple rounds. It is uncertain whether this approach saves time or computing resources compared to training a single FNO for one run on the full trajectory.

The paper also claims the cost of data acquisition from numerical solvers is extremely high. This is not well supported. According to a recent article [1], it's common for the research in the machine learning community to compare to a numerical method that is much less efficient than a SOAT method for that problem. Some of the PDE datasets are merely solved with fundamental, manually implemented Python code instead of SOAT algorithm or well-optimized software. The authors should justify that the numerical solvers used in this paper are not necessarily the best but are at least reasonably close to the SOAT solution.

Also, numerical algorithms can adjust the solving iterations to produce less accurate solutions with faster speed. As saving the cost of generating datasets for training surrogate models is the main goal of this work, it's advisable to test generating less accurate training samples from numerical solvers and compare the final accuracy and the time and computing resources cost.


[1] McGreivy, Nick, and Ammar Hakim. "Weak baselines and reporting biases lead to overoptimism in machine learning for fluid-related partial differential equations." Nature Machine Intelligence 6.10 (2024): 1256-1269.

**Essential References Not Discussed:**

No essential references I can think of.

**Experimental Designs Or Analyses:**

The experimental design was generally sound. However, one concern is that the FNO is trained using a one-input, one-output autoregressive scheme. Many previous works use a multi-step input-output approach (e.g., 10-in-10-out). While the one-step approach might achieve higher accuracy, it sacrifices prediction speed by a factor of 10. A key efficiency advantage of machine learning models over numerical solvers is they can make multi-step or skip-step predictions. The paper does not evaluate whether the proposed framework remains effective under a common multi-step configuration.

**Methods And Evaluation Criteria:**

The proposed methods and evaluation criteria make sense for the problem of surrogate modeling for PDEs.

**Other Comments Or Suggestions:**

I suggest that the authors include detailed information on computing resources and time costs for (1) the numerical solver, (2) the entire active learning procedure, and (3) the non-active learning approach. This will help readers determine whether to use active learning in their specific situations. For example, some may have a few GPUs but enough CPU cores and numerical algorithms that can effectively utilize these cores in parallel.

**Other Strengths And Weaknesses:**

The concept of using selective time steps in data is interesting. The visualization of sample diversity is clear and effectively demonstrates the motivation.

**Questions For Authors:**

The data selection method in the framework appears to be model-specific. If FNO were replaced with a different model, given the different inductive biases, would the selected time steps change? Is it possible to develop a data selection strategy that is not model-specific so that one can compare different machine learning models without requiring each to perform its own active learning process?

In my experience, for complex physical problems, predictions from models like FNO often do not fully satisfy physical constraints. Even if these predictions are fed back into the solver, it's hard to reduce the physical residuals for just a single-step solving. Will this lead to a "garbage in, garbage out" scenario where, most of the time steps selected by active learning don't satisfy physics?

**Relation To Broader Scientific Literature:**

The paper is related to AI for PDE and AI for CFD research. No apparent relevance to broader scientific literature.

**Theoretical Claims:**

No proof is provided in the paper.

---

> ### Author Rebuttal · Authors · 2025-04-01
>
> We appreciate your thoughtful questions and feedback.
>
> > Concerns about the lack of a direct comparison of computing resources with non-active learning
>
> Our paper does not claim direct computational speedups on our benchmark PDEs; instead, it relies on the benchmark PDEs as proxies reflecting realistic, expensive simulations, a common approach in active learning studies such as AL4PDE [1].
> Our surrogate metric, the number of numerical solver-simulated timesteps, effectively represents relative computational savings in realistic, expensive simulations. A computing resources analysis would be potentially misleading due to the inherently simplified nature of our experimental datasets.
>
> https://anonymous.4open.science/api/repo/icml_rebuttal-E9AB/file/taylor-green.png?v=50ac6a75
>
> Many real-world PDEs inherently demand significant computational resources for numerical simulations [2][3][4]. For instance, the Taylor-Green vortex benchmark reported in [2]—standardized through the "taubench" work unit measure—requires between $10^4$ and $10^6$ work units per trajectory, translating to roughly 41 CPU-hours to over 4,100 CPU-hours on Dual 20-Core Intel Xeon processors (Gold 6148) [5].
> This benchmark was conducted as a competition, in which multiple teams submitted numerical solvers. Each solver could run at various fidelity levels, with lower fidelity solutions requiring fewer computational resources at the cost of increased error tolerance. As shown in the image attached above (Figure 25(b) of [2]), even the fastest solution with the highest allowed error tolerance required approximately $10^4$ work units.
> This substantial computational burden highlights the importance and practicality of active learning for PDEs.
>
> > Experiments with multi-step neural solvers
>
> Thank you for raising this important point. Please see Common Response 1 at the bottom.
>
> > model-specific-ness of the method
>
> Our method STAP is model-agnostic, as it is applicable to any model that predicts the next PDE state given the current state (e.g. UNet). As you point out, STAP will likely select different timesteps when used with a different model. This is the desired behavior of any active learning algorithm, since its goal is to select data that are most useful to the current model at hand. Could you elaborate on why one would want an active learning strategy that selects the same data regardless of the model?
>
> > OOD data caused by surrogate models that potentially deviate from correct physical behavior
>
> Thank you for raising an important point. Please see Common Response 2 in our response to reviewer f7JP.
>
> **Common Response 1. Multi-step model**
>
> https://anonymous.4open.science/api/repo/icml_rebuttal-E9AB/file/multistep.png?v=ab3f6da0
>
> We have done experiments with multi-step FNO which receives N timesteps as input and outputs N timesteps. To perform STAP, we group the total number of timesteps into non-overlapping clusters of N timesteps, and perform STAP as if each cluster is one timestep with N channels.
> We have performed two variants of experiments. In the first, we divide a timestep in our main experiment into 8 smaller timesteps, so that a total of $L$ timesteps turn into $8 L$ timesteps, and train 8-in-8-out models. In the second variant, we keep $L$ timesteps the same but train 2-in-2-out models. The figure above shows the log RMSE for Burgers, KdV, and KS of the first variant, and KS, INS and CNS of the second variant.
>
> |Equation|Random|SBAL|SBAL+STAP|
> |-|-|-|-|
> |Burgers 8L/8|$-1.670\pm 0.0982$|$-1.893\pm 0.053$|$-2.058\pm 0.028$|
> |KdV 8L/8|$1.402\pm 0.029$|$1.404\pm 0.024$|$1.364\pm 0.043$|
> |KS 8L/8|$1.255\pm 0.015$|$1.232\pm 0.012$|$1.156\pm 0.008$|
> |KS L/2|$1.340\pm 0.014$|$1.335\pm 0.011$|$1.288\pm 0.011$|
> |INS L/2|$1.124\pm 0.017$|$1.118\pm 0.007$|$1.081\pm 0.012$|
> |CNS L/2|$3.593\pm 0.023$|$3.594\pm 0.044$|$3.42\pm 0.050$|
>
> The table above summarizes our results with mean log RMSE.
>
> References:
>
> [1] Musekamp, Daniel, et al. "Active learning for neural pde solvers." arXiv preprint arXiv:2408.01536 (2024). \
> [2] Wang, Q., Fidkowski, K., Abgrall, R., Bassi, F., Caraeni, D., Cary, A., ... & Olivier, H. (2012). “High-order CFD methods: current status and perspective.” International Journal for Numerical Methods in Fluids, 93(4), 212–232. \
> [3] Kaneda, Y., Ishihara, T., Yokokawa, M., Itakura, K., & Uno, A. (2003). “Energy dissipation rate and energy spectrum in high resolution direct numerical simulations of turbulence in a periodic box.” Physics of Fluids, 15(2), L21–L24. \
> [4] Heil, M. & Hazel, A. L. (2011). “Fluid–Structure Interaction in Internal Physiological Flows.” Annual Review of Fluid Mechanics, 43, 141–162.\
> [5] Capuano, F., Beratis, N., Zhang,F., Peet, Y., Squires, K., & Balaras., E. (2023). Cost vs Accuracy: DNS of turbulent flow over a sphere using structured immersed-boundary, unstructured finite-volume, and spectral-element methods. Eur. J. Mech. B Fluids, 102:91–102.

---

> > ### Comment · Reviewer_FkHn · 2025-04-06
> >
> > I just realized that the authors cannot view my official comments. I am repeating my comment here in this rebuttal comment.
> >
> > I thank the authors for their response. However, my main concern regarding the total time and computational cost remains unaddressed.
> >
> > Developing a deep learning surrogate for solving PDEs involves two major phases: dataset generation and model training. While active learning can reduce the cost of data generation, it can also increase the cost of model training. After reading the paper, I still cannot estimate whether active learning leads to net savings in total time or computational resources, as no convincing quantitative evidence is provided.
> >
> > The paper claims that the cost of generating datasets is so extensive that it dominates the total cost. I find this claim questionable because the data generation approach used in the paper may be based on computationally inefficient baselines. Please refer to my original comments. If a well-optimized numerical solver is used with appropriate PDE residual tolerance settings, the data generation cost could be significantly reduced. In such a scenario, active learning may not provide a net benefit.
> >
> > I do not expect the proposed active learning approach to always yield a net reduction in total cost. However, the authors should provide compelling quantitative evidence to clarify the conditions, such as the PDE problem's complexity, the numerical solver's efficiency, and the availability of computing resources, under which active learning can meaningfully reduce total cost.

---

> > > ### Author Response · Authors · 2025-04-08
> > >
> > > We sincerely appreciate the reviewer's reply to our response.
> > >
> > > To clarify once again, we are **not claiming a reduction of total cost in any of our benchmark PDEs**, but in hypothetical settings where the data acquisition cost far outweighs the training cost. We had thus only reported the data acquisition cost in our paper. This practice is **consistent with the broader AL literature**, including AL4PDE [1], which our work builds upon.
> > >
> > > Nonetheless, we agree that providing the total cost on our benchmarks would serve as a helpful guide to practitioners. We compare the wall clock time of non-AL and AL methods, and for a meaningful comparison, we cut the AL experiment when it reaches below the RMSE of the final non-AL surrogate model. We also note that up to our knowledge, our numerical solvers listed in Appendix B.1 are nearly SOTA in terms of computational efficiency, except that the computational cost can be further reduced by allowing higher error tolerances for some solvers (Burgers, KdV, KS and INS).
> > >
> > > **Table 1. Total wall clock time until target RMSE, in seconds**
> > > |Equation|Random|SBAL+STAP|
> > > |-|-|-|
> > > |Burgers|237|441|
> > > |KdV|834|1469|
> > > |KS|385|2920|
> > > |INS|526|3702|
> > > |CNS|4053|3476|
> > >
> > > **Table 2. Total wall clock time decomposed into acquisition/training/selection**
> > > |Equation|Random|SBAL+STAP|
> > > |-|-|-|
> > > |Burgers|90/147/0|27/234/180|
> > > |KdV|670/164/0|455/664/350|
> > > |KS |40/345/0|35/2075/810|
> > > |INS|190/336/0|160/1750/1792|
> > > |CNS|3570/483/0|1448/972/1056|
> > >
> > > Table 1 compares the wall clock times between non-AL and AL, and Table 2 decomposes them into data acquisition, model training, and data selection. We find that AL reduces the total cost in CNS, where acquisition is relatively expensive. On other benchmarks, the training and data selection costs dominate, as expected. We want to stress yet again that the benchmarks were intentionally chosen to be inexpensive, to enable fast experimentation. For instance, we lowered the CNS resolution of 256x256 in AL4PDE to 32x32, reducing the acquisition time by **around x30 times**. Since the CNS solver doesn’t explicitly set the tolerance, we empirically measured the error scale by comparing solutions from the two resolutions, which yielded an error scale of around 1e+0. This means that our numerical solver at low resolution is **already sacrificing accuracy for fast computation**.
> > >
> > > Even in settings where training cost is comparable to acquisition cost, **practical strategies** can be employed, such as using less training compute during intermediate AL rounds [2][3]. In Section 5.6 of our paper, we have also discussed methods that can significantly reduce the cost of data selection of STAP while maintaining its performance.
> > >
> > > Finally, we would like to emphasize that the reviewer’s concern applies broadly to the active learning of PDEs as a whole. We ask that the reviewer also judges our work based on its novelty in the scope of active learning for PDEs.
> > >
> > > References:\
> > > [1] Musekamp, Daniel, et al. "Active learning for neural pde solvers." arXiv preprint arXiv:2408.01536 (2024).\
> > > [2] Coleman, Cody, et al. "Selection via proxy: Efficient data selection for deep learning." arXiv preprint arXiv:1906.11829 (2019).\
> > > [3] Jung, Seohyeon, Sanghyun Kim, and Juho Lee. "A simple yet powerful deep active learning with snapshots ensembles." The Eleventh International Conference on Learning Representations. 2022.
> > >
> > > ## Addendum
> > >
> > > We provide the condition under which AL reduces total cost. Suppose AL improves data efficiency by $E$ over non-AL. Define $ T_{\text{acquire}}, T_{\text{train}} $ as the acquisition time and training time per unit data, and $T_{\text{select}}$ as the data selection time per round. The total cost of non-AL is
> > >
> > > $$ N_{\text{acquire}}^{(1)}T_{\text{acquire}} + N_{\text{train}}^{(1)}T_{\text{train}} $$
> > >
> > > and for AL,
> > >
> > > $$N_{\text{acquire}}^{(2)}T_{\text{acquire}} + N_{\text{train}}^{(2)}T_{\text{train}} + M T_{\text{select}} $$
> > >
> > > where $ N_{\text{acquire}}^{(i)} $ are the number of acquired data, and $ N_{\text{train}}^{(i)} $ are the total number of training examples (counting duplicates), and $M$ the number of rounds. With initial datasize $D$ and acquired datasize $B$ per round,
> > > $$N_{\text{acquire}}^{(1)} =BM$$
> > > $$ N_{\text{train}}^{(1)} = D+BM$$
> > > $$N_{\text{acquire}}^{(2)} = BM/E$$
> > > $$ N_{\text{train}}^{(2)}=\sum_{\text{round}=0}^{M/E} (D + B\cdot \text{round}) $$
> > >
> > > For AL to reduce the total cost, the setting would need to satisfy
> > >
> > > $$ N_{\text{acquire}}^{(1)}T_{\text{acquire}} + N_{\text{train}}^{(1)}T_{\text{train}} > N_{\text{acquire}}^{(2)}T_{\text{acquire}} + N_{\text{train}}^{(2)}T_{\text{train}}+ M T_{\text{select}}$$
> > >
> > > **Table 3. Variables for cost analysis**
> > > |Equation|$E$|$T_\text{acquire}$|$T_\text{train}$|$T_{\text{select}}$|Satisfied|
> > > |-|-|-|-|-|-|
> > > |Burgers|3.33|0.087|0.101|60|F|
> > > |KdV|1.43|0.654|0.106|50 |F|
> > > |KS|1.11|0.005|0.116|90|F|
> > > |INS|1.25|0.077|0.112|224|F|
> > > |CNS|2.5|1.760|0.157|264|T|
> > >
> > > Table 3 lists these values, and whether they satisfy the condition above.

---

### Official Review · Reviewer_vQAV · 2025-03-14

**Overall Recommendation:** 2

**Summary:**

This paper develops an acquisition function that estimates the utility of a set of time steps and utilizes it for batch active learning in training surrogate models for PDEs. The empirical results show that the proposed method outperforms the baselines.

**Claims And Evidence:**

The proposed algorithm relies on the heuristic of replacing simulations with predictions from the surrogate model for the skipped time steps. However, especially in the early stages, when the surrogate model may perform poorly, the sequence after the skipped time steps might not correlate with the initial condition u_0 at all. In such cases, why not query more diverse initial conditions and run fewer time steps for each scenario using the ground-truth simulator?

**Essential References Not Discussed:**

The related work has been thoroughly discussed.

**Experimental Designs Or Analyses:**

I am not convinced by the authors' claim that their approach improves performance by up to five times compared to the baselines. For instance, in Figure 3, when comparing SBAL and SBAL+STAP, their performance appears similar. In most cases, SBAL is only one iteration behind SBAL+STAP. Additionally, it would be helpful if the authors can include more iterations to show the full performance until convergence. In Table 2, the results indicate that random time step selection outperforms STAP on 2 out of 5 tasks.

**Methods And Evaluation Criteria:**

The evaluation metrics are well-established. For benchmarks, it is helpful to test whether the proposed methods can also be applied to PDEs where the evolution operator is time-dependent.

**Other Comments Or Suggestions:**

Please refer to the previous sections.

**Other Strengths And Weaknesses:**

Please refer to the previous sections.

**Questions For Authors:**

Please refer to the previous sections.

**Relation To Broader Scientific Literature:**

There is a amount of work on batch active learning in various scenarios. This paper focuses on actively selecting the time stamps within each scenario to further improve the sample efficiency.

**Theoretical Claims:**

There is no theoretical guarantee provided for the proposed acquisition function. It will be helpful if the authors could include a theoretical analysis of their designed acquisition function. For instance, can they prove that the proposed acquisition function provides an optimal or near-optimal solution?

---

> ### Author Rebuttal · Authors · 2025-04-01
>
> We appreciate your thoughtful questions and feedback.
>
> > Theoretical analysis of the acquisition function
>
> Our acquisition function is an approximation to the expected error reduction (EER), which is statistically near-optimal for active learning [1][2]. The EER measures how much the model’s generalization error is likely reduced after updating on hypothetically acquired data.  We model our hypothetical belief about the ground truth solver as a *uniform categorical distribution* over the ensemble $ \\{\hat{G}\_a\\}\_{a=1}^M $. We assume that acquiring the trajectory of $u^0$ with sampling pattern $S$ only reduces generalization error on the trajectory of $u^0$. The current generalization error is expected to be the average of $  \| \hat u\_a - \hat u\_b \|^2  $ over $b$. We make a second assumption that the hypothetically acquired data $ \hat{u}\_{b,S,a} $ will update the model such that the model predicts the trajectory $ \hat{u}\_{b,S,a} $ given $u^0$. This gives us the expected reduction in error $ \| \hat u\_a - \hat u\_{b} \|^2 - \| \hat u\_a - \hat u\_{b,S,a} \|^2 $ averaged across $a$ and $b$, which is equal to our acquisition function. Although proving an optimality bound is outside of our expertise, we emphasize that our design of the acquisition function was guided by this exact theoretical consideration.
>
> > OOD data due to poor quality of surrogate models in early stages
>
> Thank you for raising an important point. Please see Common Response 2 in our response to reviewer f7JP.
>
> > Performance metrics
>
> https://anonymous.4open.science/api/repo/icml_rebuttal-E9AB/file/diff.png?v=0041f607
>
> The linked image shows the improvement in log RMSE over Random (no active learning). In the KS equation, SBAL+STAP improves over Random five times as much compared to SBAL.
>
> |Equation|Random|SBAL|SBAL+STAP|
> |-|-|-|-|
> |Burgers|0.1522|0.2277|0.2451|
> |KdV|0.0831|0.1052|0.1132|
> |KS|0.1001|0.1031|0.1094|
> |INS|0.0806|0.0869|0.0912|
> |CNS|0.0544|0.0795|0.0884|
>
> However, we agree that this quantity might be misleading to readers, and thus provide a table of the measure of data efficiency, defined as the average reduction in log RMSE per round. Overall, SBAL+STAP improves data efficiency by about 10 percent compared to SBAL, which is the SOTA algorithm. For a simulation that takes ten days to run, this would amount to saving a whole day. We encourage you to look at the results in Figure 4 of Musekamp et al. [3], Figure 3(b) of Li et al. [4] (the yellow and blue lines correspond to no active learning and active learning), and Figure 5 of Bajracharya et al. [5]. Our method provides arguably the largest and the most robust performance gain reported in the PDE active learning literature.
>
> > random time step selection outperforms STAP
>
> As shown in Table 2, no single choice of $p$ for random time step selection outperforms full trajectory sampling on all PDE datasets, sometimes even degrading the performance. Unless there is a way to adaptively select $p$, random time step selection is not a viable active learning method, and was used in our paper solely for the purpose of analysis.
>
> > Experiments with more iterations
>
> https://anonymous.4open.science/api/repo/icml_rebuttal-E9AB/file/20_rounds.png?v=7ac767fc
>
> We have performed the main experiment for 20 rounds instead of 10, on Random, SBAL, and SBAL+STAP. We observe that the gap between SBAL and SBAL+STAP keeps widening, except in the KdV equation.
>
> > Experiments with time-dependent PDEs
>
> https://anonymous.4open.science/api/repo/icml_rebuttal-E9AB/file/time_dependent_ins.png?v=f9c179ae
>
> |Equation|Random|SBAL|SBAL+STAP|
> |-|-|-|-|
> |Time-dependent INS|$-0.080\pm 0.011$|$-0.081\pm 0.013$|$-0.339\pm 0.015$|
>
> We have performed an experiment on a time-dependent incompressible Navier Stokes equation, simply by using the time-dependent external force in our current INS equation. The new forcing term is a sinusoidal mixture of two spatial coordinates and the temporal coordinate. The above figure and table summarize the log RMSE of Random, SBAL, and SBAL+STAP. **Our method aligns closely with time-dependent PDEs, explaining the massive gain in performance.**
>
> References:
>
> [1] Settles, Burr. "Active learning literature survey." (2009). \
> [2] Roy, Nicholas, and Andrew McCallum. "Toward optimal active learning through sampling estimation of error reduction." ICML. Vol. 1. No. 3. 2001. \
> [3] Musekamp, Daniel, et al. "Active learning for neural pde solvers." arXiv preprint arXiv:2408.01536 (2024). \
> [4] Shibo Li, Xin Yu, Wei Xing, Robert Kirby, Akil Narayan, and Shandian Zhe. Multi-resolution active learning of fourier neural operators. In International Conference on Artificial Intelligence and Statistics, pp. 2440–2448. PMLR, 2024. \
> [5] Pradeep Bajracharya, Javier Quetzalcóatl Toledo-Marín, Geoffrey Fox, Shantenu Jha, and Linwei Wang. Feasibility study on active learning of smart surrogates for scientific simulations. arXiv preprint arXiv:2407.07674, 2024.

---

### Decision · Program_Chairs · 2025-05-01

**Decision:**

Accept (poster)

**Comment:**

This paper introduces an active learning framework that improves sample efficiency in PDE surrogate modeling by selectively acquiring only the most informative time steps rather than the full trajectories. The method introduces a variance-reduction-based acquisition function and is evaluated on several PDE benchmarks. Results demonstrate that the method can be used as a promising add-on to existing active learning strategies.

The reviewers acknowledged the novelty and practicality of the proposed approach. They also appreciated the clear writing and thorough experiments. However, all reviewers raised concerns that should be addressed before the camera-ready version. These include the extent to which the method actually outperform existing methods, the lack of theoretical explanation for the acquisition function, and an explanation of the overall computational savings.

Given the reviews, I am on the fence about this submission and am leaning towards acceptance.